# Identification of ligand-specific G protein-coupled receptor states and prediction of downstream efficacy via data-driven modeling

Oliver Fleetwood[1], Jens Carlsson[2], Lucie Delemotte[1]*

[1]Science for Life Laboratory, Department of Applied Physics, KTH Royal Institute of Technology, Stockholm, Sweden; [2]Science for Life Laboratory, Department of Cell and Molecular Biology, Uppsala University, Uppsala, Sweden

**Abstract** Ligand binding stabilizes different G protein-coupled receptor states via a complex allosteric process that is not completely understood. Here, we have derived free energy landscapes describing activation of the $\beta_2$ adrenergic receptor bound to ligands with different efficacy profiles using enhanced sampling molecular dynamics simulations. These reveal shifts toward active-like states at the Gprotein-binding site for receptors bound to partial and full agonists, and that the ligands modulate the conformational ensemble of the receptor by tuning protein microswitches. We indeed find an excellent correlation between the conformation of the microswitches close to the ligand binding site and in the transmembrane region and experimentally reported cyclic adenosine monophosphate signaling responses. Dimensionality reduction further reveals the similarity between the unique conformational states induced by different ligands, and examining the output of classifiers highlights two distant hotspots governing agonism on transmembrane helices 5 and 7.

*For correspondence:
lucie.delemotte@scilifelab.se

## Introduction

G protein-coupled receptors (GPCRs) are membrane proteins which activate cellular signaling in response to extracellular stimuli. This process is controlled by extracellular ligands such as hormones and neurotransmitters, and the binding of these increases the probability of activating intracellular partners. GPCRs are vital in many physiological processes and constitute the most common class of drug targets (*Hauser et al., 2017*).

Much of the current understanding of GPCR signaling at the molecular level can be attributed to the progress in GPCR structure determination during the last decade (*Cherezov et al., 2007*; *Hanson et al., 2008*; *Masureel et al., 2018*; *Rasmussen et al., 2011a*; *Ring et al., 2013*; *Wacker et al., 2010*). GPCRs interconvert between inactive (R) and active (R*) states, which control G protein binding to a conserved intracellular domain via conformational rearrangements among the seven transmembrane (TM) helices (*Figure 1a*; *Manglik and Kruse, 2017*). In the absence of a bound agonist, this process is called basal activity. Ligands can bind to the orthosteric site in the receptor's extracellular domain and thereby control conformational rearrangements. Orthosteric ligands are traditionally classified as either agonists, which promote activation, antagonists, which bind to the orthosteric site but do not alter basal activity, or inverse agonists that also reduce basal activity. However, this classical view of ligand efficacy is complicated by the fact that GPCRs can signal via several intracellular partners; for example, G-proteins or β-arrestins. Most agonists will activate several signaling pathways, but agonists with the ability to activate one specific intracellular partner have also been identified, a phenomenon referred to as *biased signaling*. Based on

spectroscopy and structure determination studies, conformational changes in the receptor govern the activation of signaling pathways (*Frei et al., 2020*; *Liu et al., 2012*; *Masureel et al., 2018*), although the underlying molecular mechanisms remain elusive. Characterization of the allosteric process guiding interactions with intracellular partners is a major challenge and can only be fully understood by using a combination of different methodologies.

The term *microswitch*, or *molecular switch,* describes local structural changes in the receptor that contribute to controlling activation and can, for example, involve side chain rotamers, movement of two domains relative to each other, or a helix twist. Two microswitches implicated in the activation of class A GPCRs are an outward displacement of the transmembrane helix 6 (TM6) and twist of the highly conserved N(7.49)P(7.50)xxY(7.53) motif (superscripts notation according to Ballesteros–Weinstein numbering; *Ballesteros and Weinstein, 1995*), which take part in the formation of the intracellular binding site. In the orthosteric site, microswitches are typically less conserved and depend on the type of ligand recognized by the receptor (*Manglik and Kruse, 2017*).

Considering the high dimensionality of a protein with over 300 interacting residues, it is difficult to identify relevant microswitches from the sequence or static experimental structures. Historically, sequence analysis and mutagenesis experiments (*Gregorio et al., 2017*; *Lamichhane et al., 2020*, *Lamichhane et al., 2015*; *Manglik et al., 2015*; *Picard et al., 2019*) have been used to characterize motifs important for signaling, but this approach may overlook the role of less conserved residues, water, and ions in ligand recognition and receptor activation (*Chen et al., 2020*). Molecular dynamics (MD) simulations can generate trajectories from experimental starting structures, capture the dynamics of all microswitches and allow us to derive the free energy landscapes governing the equilibrium between protein states. MD simulations have indeed been used extensively over the last decade to study GPCR activation (*Bhattacharya and Vaidehi, 2010*; *Dror et al., 2011*; *Hu et al., 2019*; *Kohlhoff et al., 2014*; *Li et al., 2013*; *Miao and McCammon, 2016*; *Niesen et al., 2011*; *Shan et al., 2012*; *Tikhonova et al., 2013*). Due to the computational cost of brute-force MD simulations, it is nearly impossible to obtain converged results without enhanced sampling methods, although the use of special purpose hardware (*Dror et al., 2011*) has pushed the boundaries of what is achievable by conventional MD simulations. Enhanced sampling strategies have emerged as an alternative, where exploration of the conformational landscape is promoted by the introduction of a bias in the simulations (*Harpole and Delemotte, 2018*). In a post-processing step, the bias can

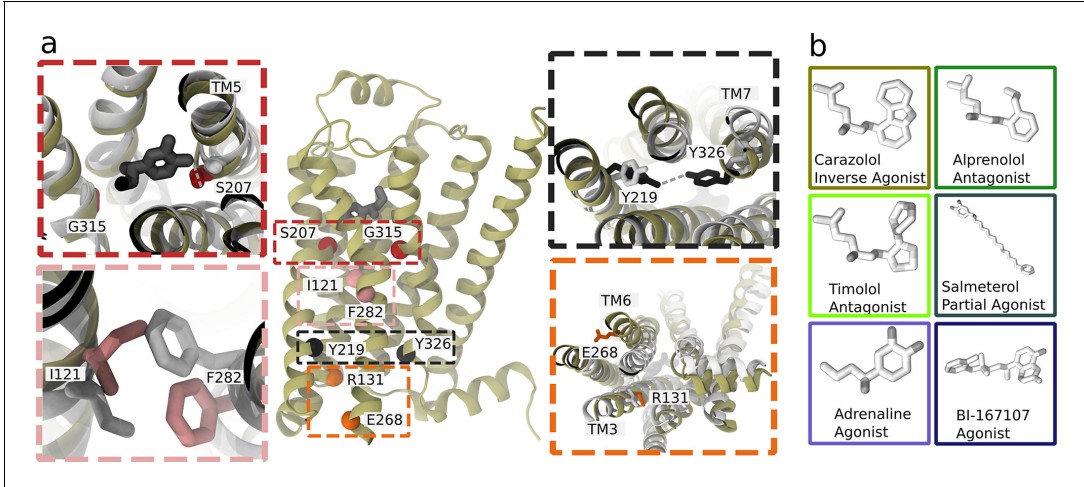

**Figure 1.** Structure and microswitches of the $\beta_2$ adrenergic receptor. (a) A molecular dynamics (MD) snapshot of the $\beta_2$ adrenergic receptor in complex with adrenaline in an active-like state (simulation starting from PDB 3P0G). The vignettes show the conformations of residue pairs reflecting important microswitches in the active and inactive structures 3P0G (color) and 2RH1 (white): the TM5 bulge (red), measured as the closest heavy atom distance between S207(5.46) and G315(7.41); the connector region's conformational change (pink), measured as the difference in root-mean-square deviation (RMSD) between the active and inactive state structure of the residues I121(3.40) and F282(6.44); the Y-Y motif (black), measured as the C-$\zeta$ distance between Y219(5.58) and Y326(7.53) of the NPxxY motif; and the Ionic lock displacement (orange), measured as the closest heavy atom distance between E268(6.30) and R131(3.50). (b) Ligands examined in this study: agonists BI-167107 and adrenaline; biased partial agonist salmeterol; antagonists timolol and alprenolol; and the inverse agonist carazolol. Atoms are colored according to their partial charge.

be adjusted for, and it is thus possible to derive theoretically exact results at a fraction of the cost of unbiased simulations. However, the ever-increasing size of simulation data and diverse conformations sampled by MD simulations makes it difficult to identify and determine the importance of microswitches by simple visualization of the conformational ensemble. Data-driven and machine learning approaches can help to condense the data and reduce human bias in the interpretation of the results (*Fleetwood et al., 2020c*; *Hu et al., 2019*).

In this study, we focus on the prototypical $\beta_2$ adrenergic receptor ($\beta_2$AR), which interacts with Gs proteins to trigger a cyclic adenosine monophosphate (cAMP) response, and arrestins, which control endocytosis and kinase activation (*Jean-Charles et al., 2017*). Both pathways are physiologically relevant and are modulated by therapeutic drugs. The $\beta_2$AR is a drug target for bronchoconstriction medication and was the first receptor crystallized in complex with a G protein (*Rasmussen et al., 2011a*; *Rasmussen et al., 2011b*). Experimental studies, including crystallography (*Masureel et al., 2018*; *Rasmussen et al., 2011b*; *Ring et al., 2013*), spectroscopy methods (*Gregorio et al., 2017*; *Imai et al., 2020*; *Kofuku et al., 2012*; *Lamichhane et al., 2020*; *Liu et al., 2012*), and computational methods (*Provasi et al., 2011*; *Chen et al., 2020*; *Dror et al., 2011*; *Kohlhoff et al., 2014*; *Tikhonova et al., 2013*), have investigated the activation mechanism of the $\beta_2$AR. Agonists bound in experimental structures show a key interaction with S207(5.46) (*Chan et al., 2016*) and an inward bulge of TM5 in the active state. In the TM domain between the orthosteric site and G protein-binding site, *the connector region* (*Weis and Kobilka, 2018*), partially overlapping with the P(5.50)I(3.40)F(6.44) motif, undergoes a rotameric change and thereby influences the hydrated cavity surrounding the conserved D79(2.50) (*Imai et al., 2020*), which in turn interacts with the conserved NPxxY motif in TM7 and reorients Y326(7.53) to form water-mediated interaction with Y219(5.58) (the Y–Y motif) (*Latorraca et al., 2017*). The combination of several microswitches leads to conformational changes that promote an outward movement of TM6 and binding of an intracellular binding partner, such as a G protein or arrestin. Understanding how ligands modulate individual microswitches could aid the development of biased agonists.

Enhanced sampling techniques have been used to characterize the activation mechanism of $\beta_2$AR, from early coarse-grained protocols (*Bhattacharya and Vaidehi, 2010*; *Niesen et al., 2011*) to more refined methodologies involving Gaussian accelerated MD (*Tikhonova et al., 2013*), metadynamics using path collective variables (CVs) derived from adiabatic biased MD simulations (*Provasi et al., 2011*), or adaptive sampling on cloud-based computing resources (*Kohlhoff et al., 2014*). Following in these footsteps, we recently introduced a version of the string with swarms of trajectories method designed to capture the activation pathway and the free energy landscapes along various microswitches (*Fleetwood et al., 2020b*).

In this study, thanks to our cost-effective computational approach, we have derived the activation free energy and characterized the details of the active-like state of the $\beta_2$AR (*Figure 1a*) in its ligand-free state and bound to six ligands with different efficacy profiles (*Figure 1b*), all of which were resolved bound to the $\beta_2$AR (*Figure 1b*) and several of which are clinically approved drugs (*Woo and Robinson, 2015*). The free energy landscapes revealed a stabilization of active-like states for the receptor bound to agonists and a shift toward inactive-like states for the receptor bound to antagonists or inverse agonists. Remarkably, we obtained a strong quantitative correlation between experimentally measured intracellular cAMP responses and the expectation values of the upper and transmembrane microswitches, highlighting the predictive power of our approach. In a second step, we introduce an adaptive sampling protocol developed to quantitatively sample the most stabilized states kinetically accessible from the activated starting structure (which we will refer to as the active-like state). Using dimensionality reduction techniques, we find that all ligands stabilize distinct receptor states and that ligands with similar pharmacological properties cluster together. Several of the microswitches considered to be of significance for GPCR activation, such as the NPxxY motif and the extracellular end of TM5, were automatically identified as important with our protocol. Combined with the activation free energies, our results show how ligands control the population of states. They modulate the conformational equilibrium by tuning important allosteric microswitches, in particular near the G protein-binding site. By inspecting the inter-residue contacts formed for different ligands, we identified an allosteric pathway between the two binding sites and a large heterogeneity of TM7 states. Our results thus build on the earlier use of enhanced sampling methods and demonstrate how such protocols combined with today's computational capacities and availability of high-resolution structures in various states can provide insights into the structural basis of allosteric

**Table 1.** Total simulation time per string of swarm simulation ensemble*.

| Ligand | Steered molecular dynamics simulation time [μs] | #Restrained equilibration trajectories (30 ps each) | #Swarm trajectories (10 ps each) | Total simulation time [μs] |
|---|---|---|---|---|
| Carazolol | 0.2 | 14878 | 352816 | 4.17 |
| Alprenolol | 0.2 | 14878 | 364856 | 4.29 |
| Timolol | 0.2 | 14878 | 363808 | 4.28 |
| Salmeterol | 0.2 | 14878 | 363232 | 4.28 |
| Adrenaline | 0.2 | 14878 | 372936 | 4.38 |

* The previously published apo and BI-167107 initiated systems followed a slightly different initialization protocol with three substrings (*Fleetwood et al., 2020b*) and have been excluded from the table.

communication and ligand efficacy profiles, and potentially find use in the design of novel GPCR drug candidates.

## Results

### Ligands control efficacy by reshaping microswitches' probability distributions

We derived the free energy landscape along the most probable activation pathway of the β₂AR bound to different ligands using the string method with swarms of trajectories (*Figure 1b*). The set of ligands studied consisted of the full agonists BI-167107 and adrenaline, the G protein-biased agonist salmeterol, the antagonists alprenolol and timolol (sometimes classified as a partial inverse agonist; *Hanson et al., 2008*), and the inverse agonist carazolol. After 305 iterations, corresponding to 4 μs of aggregated simulation time per ligand-receptor complex, the activation pathways had converged (*Table 1* and *Figure 2—figure supplements 1–6*). Based on the short swarm trajectories, we calculated free energy landscapes along different microswitches identified previously (*Fleetwood et al., 2020b*; *Figure 2a,b* and *Figure 2—figure supplement 7*): (1) the TM5 bulge, which is an indicator of contraction in the ligand binding site; (2) the connector ΔRMSD, a rotameric switch involving residues I121(3.40) and F282(6.44) in the TM region; (3) the ionic lock distance reflecting the outward movement of TM6, measured as the closest heavy atom distance between E268(6.30) and R131(3.50); and (4) the Y-Y motif as the C-ζ distance between Y219(5.58) and Y326 (7.53), which acts as an indicator of the twist of the NPxxY motif and a slight reorientation of TM5.

The free energy landscapes projected along the connector ΔRMSD (*Figure 2a*) reveal two states. In agreement with what could be expected, agonists lower the relative free energy of the active state (R*) of this microswitch, whereas non-agonists favor the inactive state (R) more. A loose coupling between the orthosteric ligand and G protein-binding site was proposed based on correlated motions between the two domains in long timescale MD simulations of the BI-167107-bound receptor (*Dror et al., 2011*). The free energy landscapes projected along the TM5 bulge in the orthosteric site and the ionic lock distance in the G protein-binding site (*Figure 2b*) provide a quantitative view of this correlation and reveal that the activation pathway and the precise conformation of the stabilized states along the pathway depends on the ligand (*Tikhonova et al., 2013*; *Kohlhoff et al., 2014*). In general, the TM5 bulge assumed an outward conformation when TM6 was in its inward, inactive state. Furthermore, non-agonists favored a conformation with both a fully inactive TM5 bulge and an inactive TM6, whereas agonists favored a more contracted binding site even in the inactive state of TM6. However, despite the relatively loose coupling, it should be noted that agonists were generally observed to shift the energy balance to favor active-intermediate receptor conformations with a TM6 displacement larger than in the inactive state.

It is not straightforward to predict ligand efficacy by visual inspection of a free energy landscape, since it is the Boltzmann integrals over the basins that determine the relative free energy of the

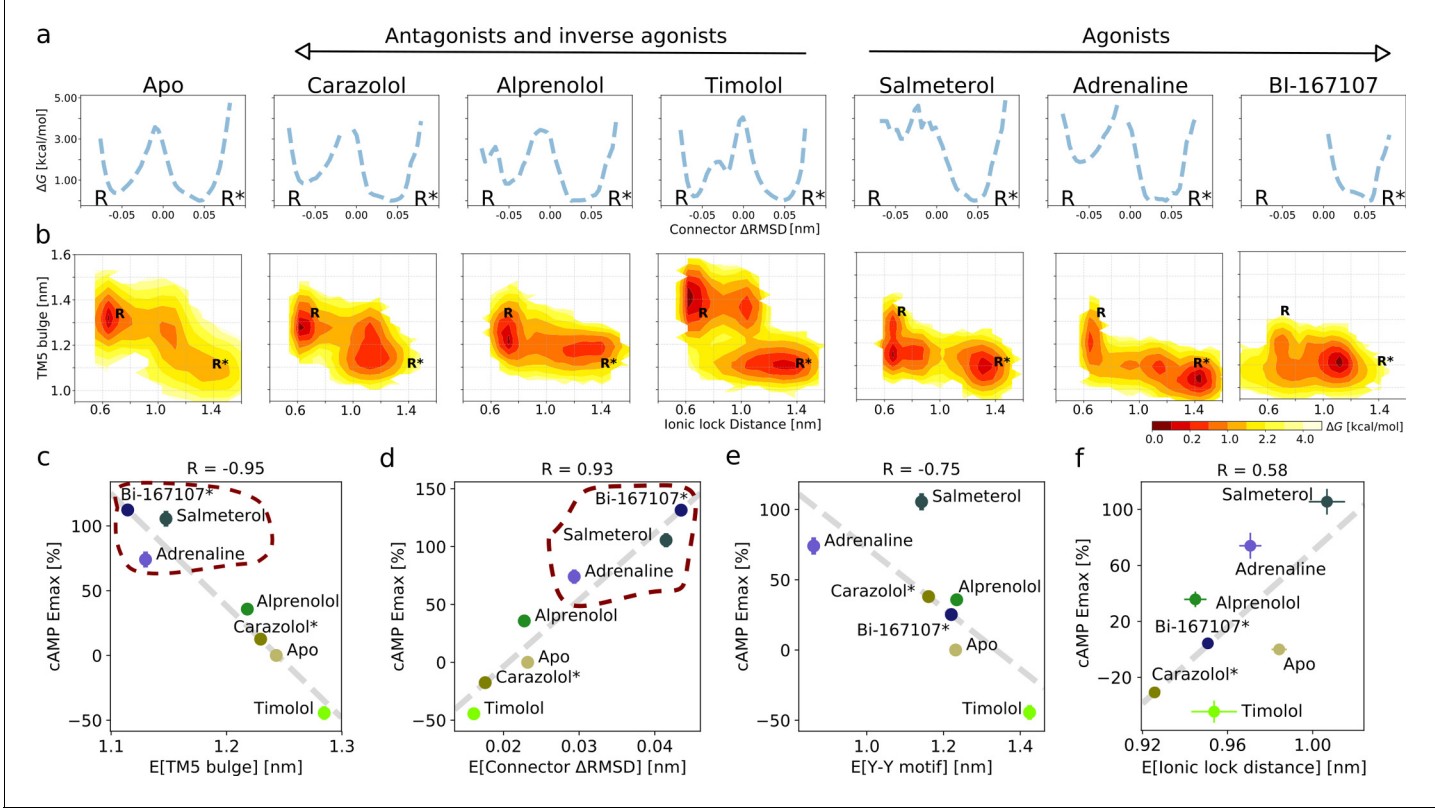

**Figure 2.** Ligand-dependent free energy landscapes and expected downstream response. (a) Free energy landscapes for the different ligands projected along the connector ΔRMSD microswitch. The inactive and active states are marked by R and R* respectively. (b) Free energy projected onto the TM5 bulge CV in the orthosteric site and the ionic lock distance (measuring TM6 displacement in the G protein-binding site). (c)–(f) Correlation between experimental values of downstream cyclic adenosine monophosphate (cAMP) signaling and the expectation value of different microswitches for the receptor bound to different ligands, (c) for the TM5 bulge, (d) for the connector ΔRMSD, (e) for the Y-Y motif, and (f) for the ionic lock distance. The cAMP Emax values for carazolol and BI-167107 (marked with an asterisk), which were not available in the experimental study, are inferred from the linear regression. Red dashed lines highlight the clustering of the agonist-bound structures.

The online version of this article includes the following figure supplement(s) for figure 2:

**Figure supplement 1.** Strings averaged over different iterations for the carazolol-bound receptor initiated from the active starting structure.

**Figure supplement 2.** Strings averaged over different iterations for the carazolol-bound receptor initiated from the inactive starting structure.

**Figure supplement 3.** Strings averaged over different iterations for the alprenolol-bound receptor initiated from the inactive starting structure.

**Figure supplement 4.** Strings averaged over different iterations for the timolol-bound receptor initiated from the inactive starting structure.

**Figure supplement 5.** Strings averaged over different iterations for the salmeterol-bound receptor initiated from the inactive starting structure.

**Figure supplement 6.** Strings averaged over different iterations for the adrenaline-bound receptor initiated from the inactive starting structure.

**Figure supplement 7.** Free energy landscapes projected along important microswitches.

**Figure supplement 8.** Correlation between experimental values of downstream cyclic adenosine monophosphate (cAMP) signaling and the relative free energy of states for the receptor bound to different ligands.

**Figure supplement 9.** Comparison between the two pathways obtained for the carazolol-bound receptor.

**Figure supplement 10.** Correlation between experimental values of downstream cyclic adenosine monophosphate (cAMP) signaling and microswitch expectation value for the receptor bound to different ligands.

**Figure supplement 11.** Free energy landscapes for the receptor in its apo state (left column), bound to carazolol (center column), and bound to BI-167107 (right column).

active and inactive states (ΔG). To investigate if ligand efficacy could be quantified using our simulation results, we computed expectation values and ΔG of the microswitches and compared them to functional experiments measuring the maximal G protein-mediated cAMP production (Emax) (*Figure 2c–f* and *Figure 2—figure supplement 8*; *van der Westhuizen et al., 2014*). Remarkably, the expectation values associated with the upper microswitches were strongly correlated to the

previously reported experimental values, in particular the TM5 bulge (*Figure 2c*; R = −0.95) and the connector ΔRMSD (*Figure 2d*; R = 0.93). Emax values of BI-167107 and carazolol were not available, and we thus inferred their predicted Emax from the linear correlation obtained for the other ligands: we predicted BI-167107 to have a cAMP Emax value slightly higher than adrenaline and salmeterol, and carazolol to have an Emax similar to the values of the ligand free receptor and inverse agonist timolol (*Figure 2c,d*). These results are in line with expectations; BI-167107 is indeed a known full agonist and carazolol an inverse agonist (*Manglik et al., 2015*; *Rasmussen et al., 2011a*; *Ueda et al., 2019*). Similar results were obtained using the free energy difference of the active and inactive states, ΔG (*Figure 2—figure supplement 8*). These results thus suggest that our simulations accurately captured the relative stability of states and should therefore be able to provide insights into how ligands with different efficacy profiles control the conformational ensemble of the receptor. Moving down the microswitch cascade toward the intracellular region, the cAMP response was less well correlated with the expectation values and the ΔG of the Y-Y motif and Ionic lock distance (*Figure 2e*; R = −0.75 and *Figure 2f*; R = 0.58, respectively, and *Figure 2—figure supplement 8*), as expected from the looser coupling between these microswitches and the ligand binding sites.

As a control, we converged the activation string for a simulation set initiated from the inactive state structure, where the starting activation pathway was sampled in the reverse direction (*Figure 2—figure supplements 1–2* and *9a*). The TM microswitch expectation values led to a similar prediction of Emax, accurately classifying carazolol as a non-agonist (*Figure 2—figure supplement 10*). Inspection of the free energy landscapes (*Figure 2—figure supplement 9b–d*), on the other hand, revealed two differences between the carazolol-bound receptors' active states obtained starting from different initial strings (*Figure 2—figure supplement 9c–d*): (1) in the 2RH1-initiated system, the intracellular domain of TM6 assumed an orientation with the ionic lock residues' side chains pointing away from each other (*Figure 2—figure supplement 10d*), although the backbone distance between TM6 and TM3 was very similar (*Figure 2—figure supplement 9d*), and (2) the 2RH1-initiated system sampled a conformation with an inactive TM5 bulge domain and active cytosolic domain (*Figure 2—figure supplement 9c*), unlike any conformation captured in experimental structures. We hypothesize that the conformation obtained starting from the inactive state could be an artifact of pulling the inverse agonist-bound receptor directly toward its unfavorable active state, without targeting metastable intermediate states along the pathway. Moving forward, we thus favor a protocol in which the receptor was pulled along a pathway identified by unbiased MD simulations, presumably closer to the most favorable converged activation pathway (*Dror et al., 2011*; *Fleetwood et al., 2020b*).

Although downstream efficacy is an important metric for drug discovery purposes, alternative methods are required to characterize the molecular basis of receptor activation. Spectroscopy experiments have proven useful for this purpose (*Gregorio et al., 2017*; *Imai et al., 2020*; *Kofuku et al., 2012*; *Manglik et al., 2015*; *Ma et al., 2020*; *Nygaard et al., 2013*; *Ueda et al., 2019*; *Weis and Kobilka, 2018*), yet they are often difficult to compare quantitatively to atomistic simulations due to chemical modifications introduced and/or complex interpretation of measured signals. [19]F-fluorine NMR and double electron-electron resonance (DEER) spectroscopy experiments have shown that the conformational ensembles of carazolol and the apo receptor have similar TM6 distance distributions (*Manglik et al., 2015*), in agreement with the similarity in their microswitch expectation values and free energy landscapes (*Figure 2b,f* and *Figure 2—figure supplement 8e*). It has also been proposed that the inactive receptor exists in two sub-states (*Manglik et al., 2015*), one with a formed and one with a broken ionic lock. Our simulations rarely captured a sub-state with the ionic lock formed, which suggest that the state with a broken ionic lock is of lower free energy, although modeling of missing residues in the cytosolic domain of TM6 and TM3 may alter the dynamics of this region (*Dror et al., 2009*). Nevertheless, the agonists stabilized a state with the side chains of the ionic lock residues pointing away from each other, while the inverse agonist carazolol favored a state with the side chains pointing toward each other, although the TM6 displacement was too large for the residues to fully form an ionic bond (*Figure 2—figure supplement 11a*). In general, the ligands stabilized active-like states of different ionic lock displacements (*Figure 2b*). As GPCRs only assume their fully active state in the presence of an intracellular binding partner—a condition not met in the simulations carried out in this work—a loose allosteric coupling between intracellular microswitches and the cellular response is expected.

**Table 2.** String simulations: collective variables.

| Residues | Importance |
|---|---|
| F223(5.62)-A271(6.33) | 1.0 |
| Q224(5.63)-K227(5.66) | 0.97 |
| F223(5.62)-L272(6.34) | 0.76 |
| I325(7.52)-R328(7.55) | 0.73 |
| F223(5.62)-K227(5.66) | 0.72 |
| A226(5.65)-K267(6.29) | 0.69 |
| V54(1.53)-C327(7.54) | 0.67 |
| L324(7.51)-R328(7.55) | 0.67 |
| A134(3.53)-Y141 | 0.66 |
| I135(3.54)-L272(6.34) | 0.6 |
| V222(5.61)-A271(6.33) | 0.59 |
| A226(5.65)-E268(6.30) | 0.59 |
| Q26(1.25)-D29(1.28) | 0.57 |
| I135(3.54)-E225(5.64) | 0.57 |
| R131(3.50)-L275(6.37) | 0.57 |
| A134(3.53)-A271(6.33) | 0.56 |
| C285(6.47)-V317(7.43) | 0.56 |
| A76(2.47)-P323(7.50) | 0.56 |
| Q26(1.25)-E30(1.29) | 0.55 |
| A226(5.65)-A271(6.33) | 0.54 |
| I121(3.40)-F208(5.47) | 0.53 |
| E338(8.56)-R343 | 0.53 |
| I334(8.52)-R344 | 0.52 |
| G50(1.49)-L324(7.51) | 0.49 |
| T25-D29(1.28) | 0.47 |
| I135(3.54)-A271(6.33) | 0.46 |
| T25-E30(1.29) | 0.46 |
| W286(6.48)-G315(7.41) | 0.45 |
| C285(6.47)-N318(7.45) | 0.44 |
| Q27(1.26)-E30(1.29) | 0.44 |
| R63-D331(8.49) | 0.43 |
| Q197(5.37)-V297(6.59) | 0.43 |
| A134(3.53)-E268(6.30) | 0.42 |
| P288(6.50)-L311(7.37) | 0.42 |
| T281(6.43)-N318(7.45) | 0.42 |
| C341(8.59)-R344 | 0.42 |
| I135(3.54)-P138 | 0.4 |
| R328(7.55)-R333(8.51) | 0.36 |
| L311(7.37)-G315(7.41) | 0.36 |
| I135(3.54)-E268(6.30) | 0.35 |
| C285(6.47)-I314(7.40) | 0.34 |

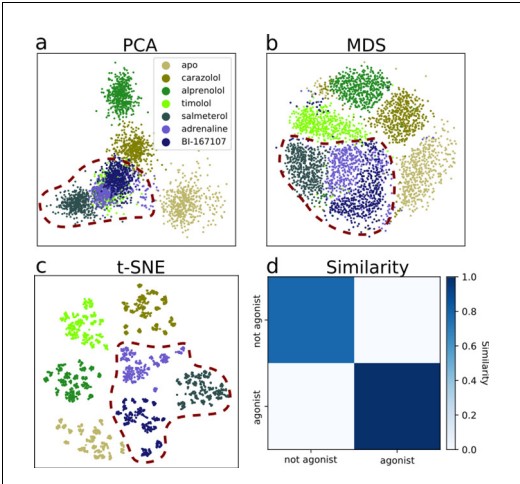

**Figure 3.** Dimensionality reduction techniques applied to the active-like simulation ensembles. Each point represents a simulation snapshot, colored according to the ligand bound to the receptor. Red dashed lines highlight regions where agonists cluster. The features are computed as the inverse closest heavy atom distances between residues. (a) Principal component analysis (PCA) projection onto the first two principal components (PCs), (b) multi-dimensional scaling (MDS) and (c) t-distributed stochastic neighbor embedding (t-SNE). (d) The similarity between conformations sampled when agonist (BI-167107, adrenaline and salmeterol) and non-agonist ligands are bound, measured as the average distance between configurations in the full feature space.

The online version of this article includes the following figure supplement(s) for figure 3:

**Figure supplement 1.** Dimensionality reduction of the activation pathways, applied to the activation paths derived from the swarms of trajectories method.

**Figure supplement 2.** Equilibration of single states.

**Figure supplement 3.** Dimensionality reduction plots and similarity metrics for the kinetically trapped active-like states, using three subsets of the simulation frames in (a)–(c).

**Figure supplement 4.** State projection onto higher-order PCs.

β$_2$AR crystal structures reveal a number of stabilized water molecules in the inactive state, while this region is dehydrated in the G protein-bound state (*Cherezov et al., 2007*; *Rasmussen et al., 2011b*). The disruption of intra-receptor water networks and the formation of a hydrophobic barrier, a prerequisite of activation (Trzaskowski et al. 2012), are likely conserved features of activation (*Venkatakrishnan et al., 2019*). Hydration in the active state may also contribute to the change in probe environment observed in spectroscopy experiments (*Lamichhane et al., 2015*). We investigated the hydration near the intracellular binding site by counting the number of water molecules within 0.8 nm of L266 (6.28) (*Figure 2—figure supplement 11b*), and found that BI-167107 stabilized a partially dehydrated active state when TM6 assumed its outward pose. Carazolol and the apo condition, on the other hand, did not induce dehydration with TM6 in its active conformation (*Figure 2—figure supplement 11b*). This finding shows that, to fully understand agonist control of GPCR activation, one needs to combine the shift in free energy with the conformational differences between the states induced by the ligands.

## Data-driven analysis reveals that ligands stabilize unique states

To pinpoint the molecular basis of signaling, we reduced the high dimensional datasets to a more compressed representation using methods from machine learning (*Figure 3*). We used three different methods to analyze all inter-residue contacts: principal component analysis (PCA), multidimensional scaling (MDS), and t-distributed stochastic neighbor embedding (t-SNE). All these approaches were designed to find a low dimensional embedding of the high dimensional data, but differ in their underpinnings. PCA seeks a linear transformation of the input data into an orthogonal basis of principal

components (PCs) and is designed to cover as much of the variance in the data as possible (*Figure 3a*). MDS projects the high dimensional space into a low dimensional representation using a non-linear transformation which preserves the distance between points (*Figure 3b*). T-SNE is a visualization technique which seeks to disentangle a high-dimensional data set by transforming it into a low-dimensional embedding where similar points are near each other and dissimilar objects have a high probability of being distant (*Figure 3c*).

We evaluated two datasets: the equilibrated active-like state ensemble (*Figure 3*) and the swarm trajectories from the final iteration of the converged string. For the latter, which represent the converged activation pathways, the dimensionality reduction techniques created embeddings which separated snapshots by their progression along the activation path (*Figure 3—figure supplement 1*). This is expected because unsupervised dimensionality reduction methods tend to emphasize large scale amplitude motions, such as the displacement of TM6 in the case of GPCR activation

(*Fleetwood et al., 2020c*). These results confirmed that this feature is shared among all activation pathways, regardless of which ligand the receptor was bound to.

Whereas the activation path ensembles contained inactive, intermediate, and active-like states for every ligand–receptor complex, the active-like state ensemble simulations revealed that the conformations sampled in the presence of different ligands differed substantially. Indeed, after eight iterations with an accumulated simulation time of 1.4 µs per ligand (see Materials and methods), the method for finding single equilibrated states generated trajectories that diffused around the most stabilized state kinetically accessible from the starting structure (*Figure 3—figure supplement 2*). Dividing the dataset into thirds yielded similar results (*Figure 3—figure supplement 3*), showing that the states were adequately sampled.

The details of the active state ensemble varied among the ligands (*Figure 3a–c*). Simulations with agonists bound tended to be grouped together for all three dimensionality reduction methods, but each of them generally also led to a distinct conformational ensemble. The simulation snapshots with the agonist adrenaline bound were generally close to the full agonist BI-167107 and the partial agonist salmeterol. For the other ligands, the receptor explored a different conformational space than with agonists, but the ensembles were more diverse. In agreement with the projections, the similarity matrix based on the average distance between snapshots in the full feature space (*Figure 3d*) showed that agonists and non-agonists stabilized significantly different states.

Using PCA, we note that timolol clusters together with the agonist ligands. Thus, the first two PCs are not sufficient to completely separate the dataset according to the ligands present (*Figure 3a*), but including more PCs in the projection leads to a satisfactory separation (*Figure 3—figure supplement 4*). The non-linear methods separated the classes well in two dimensions. As expected, a few points deviated from the other snapshots in the same class due to sampling slightly outside defined free energy basins. We also note that although t-SNE generates an embedding with perfect separation between classes, the micro-clusters depend on parametrization of the method and their exact placement is stochastic (*Schubert, 2017*).

To summarize, an analysis of the simulations by machine learning shows that ligands share many overall features of activation, but stabilize unique local states, in agreement with previous work (*Kohlhoff et al., 2014*; *Tikhonova et al., 2013*; *Provasi et al., 2011*; *Liu et al., 2012*; *Lamichhane et al., 2020*; *Frei et al., 2020*; *Suomivuori et al., 2020*). Together with the free energy landscapes, our findings support that ligands control the relative time a receptor spends in active-like states, and induces small conformational state-specific signatures throughout the protein.

## Ligands control residues near the G protein-binding site

To capture the important characteristics of receptor activation, we applied PCA on the swarms of trajectories datasets representing the activation ensemble and extracted important features from these (*Figure 4c*). This analysis identified parts of TM6 and TM7 near the G protein-binding site as particularly important (*Figure 4c* and *Figure 4—figure supplement 1*), adding further support for the importance of these microswitches for activation. To characterize molecular differences between the active-like states controlled by the different ligands, we applied supervised learning on our dataset. With this approach, we derived features discriminating between the classes based on inter-residue distances and thereby identified residues which could be important for activation. Importance profiles were computed for discriminating between agonists and non-agonists (*Figure 4a*) and to distinguish all ligands from each other (*Figure 4b*) using a symmetrized version of the Kullback–Leibler (KL) divergence (*Fleetwood et al., 2020c*; *Kullback and Leibler, 1951*). With this approach, two residues constituting a distance were scored as important if the active-like states formed non-overlapping distance distributions, corresponding to a high KL divergence. As a control, we also evaluated the important features learned by a random forest (RF) classifier, a machine learning classifier constructed by an ensemble of decision trees. The importance profiles of the KL and RF feature extractors were similar, although the RF classifier generated importance profiles with more distinct peaks. Since the datasets included a few simulation frames that fluctuated outside the equilibrated states, the RF classifier probably suppressed some features to enhance prediction accuracy for these frames. KL divergence estimated how much the distributions overlapped along individual features and was therefore less likely to discard features based on these frames. Remarkably, both data-

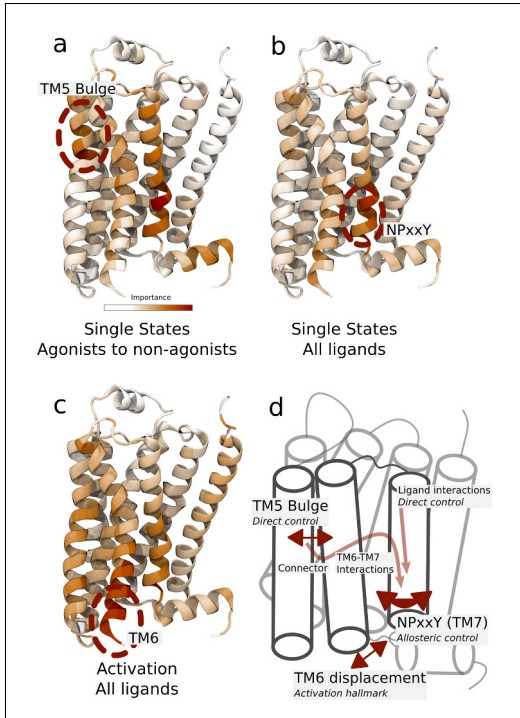

**Figure 4.** Important residues for distinguishing ligand-dependent activation mechanisms. (**a**)–(**b**) Residues identified to be important for classification of the equilibrated active-like states. Importance was derived by computing the Kullback–Leibler (KL) divergence along all features, followed by averaging per residue. (**a**) Comparison of agonists and non-agonists. One signaling hotspot is located at the transmembrane 5 (TM5) bulge and another on TM7 close to the NPxxY motif. (**b**) Important residues to discriminate between all ligands. The main hotspot is located near the NPxxY motif. (**c**) Important residues for the activation ensemble from the swarms of trajectories method, extracted with PCA. The importance per feature was computed as the product of the PC's weights and the PC's projection onto the input feature. The intracellular end of TM6, which undergoes a large conformational change upon activation, is marked as important. Inverse closest heavy atom distances were used as input features in all figures. (**d**) Conceptual model describing allosteric communication between the hotspots. Ligands exercise direct control of TM5, which is stabilized in different states by agonists and non-agonists. In turn, residues approximately one helical turn below the orthosteric site, including the connector region, couple to the conformations in the orthosteric site. This leads to distinct interaction patterns between TM6 and TM7 in the TM domain. The importance of TM7 is further enhanced by direct ligand interactions. By favoring distinct TM7 states and modulating the probability of stabilizing TM6 in an active conformation, ligands control the G protein-binding site.

*Figure 4 continued on next page*

driven methods identified established micro-switches as the most important regions for classification: the NPxxY motif and the intracellular part of TM6 in the intracellular binding site (*Figure 4a,b* and *Figure 4—figure supplement 2*).

These results underpin experimental evidence that differences in these regions are related to biased signaling (*Frei et al., 2020*; *Lamichhane et al., 2020*, *Lamichhane et al., 2015*; *Liu et al., 2012*; *Suomivuori et al., 2020*). The TM5 bulge was particularly important for discriminating between agonists and non-agonists (*Figure 4a*). This region does not show up as important when differentiating between all ligands (*Figure 4b*), which means that the TM5 conformations within the two groups of ligands were so similar that this region could not be used to, for example, discriminate agonists from each other. The NPxxY motif, on the other hand, assumed a unique conformation for each ligand (*Figure 4b*).

Spectroscopy experiments have shown that the agonist BI-167107 stabilizes an intermediate, pre-active state (*Manglik et al., 2015*). It was hypothesized that receptor activation involves a transition via this state before forming the fully active state together with an intracellular binding partner (*Manglik et al., 2015*). The experimental response was too weak to discern a corresponding pre-active state for antagonists, but the authors found it likely that such a state is accessible to all ligands. Moreover, spectroscopy experiments found that different agonists induced different states in the cytoplasmic domain (*Manglik et al., 2015*). Our results provide molecular models for the pre-active ensemble, and identified the region around the NPxxY domain a major source of conformational heterogeneity (*Figure 4b* and *Figure 4—figure supplements 1–4*). In agreement, conformational differences in this domain have been shown to correlate to efficacy and biased signaling in [19]F NMR and single-molecule fluorescence spectroscopy experiments (*Frei et al., 2020*; *Lamichhane et al., 2020*; *Liu et al., 2012*).

Taken together, we arrived at a conceptual model to describe how different ligands control the G protein-binding site (*Figure 4d*). Ligands exercise direct control of TM5, where agonists and non-agonists stabilize different states. In turn, residues approximately one helical turn below the orthosteric site—including the connector region, which was identified as a good predictor of downstream response

*Figure 4 continued*

The online version of this article includes the following figure supplement(s) for figure 4:

**Figure supplement 1.** Feature importance projected onto snakeplots.

**Figure supplement 2.** Comparison between supervised feature extraction methods.

**Figure supplement 3.** Important activation features per ligand identified by applying unsupervised principal component analyis (PCA) on the activation paths.

**Figure supplement 4.** Important features for equilibrated active-like states.

(*Figure 2d*)—couple to the conformations in the orthosteric site. This leads to distinct interaction patterns between TM6 and TM7 in the TM domain. The importance of TM7 is further enhanced by direct ligand interactions, generating a variety of ligand-specific NPxxY motifs. By favoring distinct TM7 states and modulating the probability for TM6 to be in an active conformation, ligands hence control the G protein-binding site. The overall pattern is compatible with observations made from an MSM analysis of large-scale computations (*Kohlhoff et al., 2014*). However, since our simulation protocol achieves conformational sampling at a fraction of the computational cost, it has allowed us to compute the free energy landscape for a larger ligand dataset and to thus find the molecular basis for the effect of binding of various agonists, antagonists, and inverse agonists.

## Molecular basis for allosteric transmission from the orthosteric to the intracellular binding site

To further explore the atomistic basis of our conceptual model (*Figure 4*), we systematically inspected the most important features connecting the two hotspots near the TM5 bulge and the NPxxY motif. As in the previous section, we computed the KL divergence of the inter-residue distance distributions between the ligands. Distances with a high KL divergence that contributed to the formation of ligand-specific active-like states were further investigated. Although the identified residue-pairs did not necessarily reflect the causality of molecular interactions driving the conformational changes, key features of activation were captured by this automated approach.

We first identified features shared between agonists near the orthosteric site. We found that V206(5.46) could form van der Waals interactions with T118(3.37) only in the presence of agonists (*Figure 5a and b*). This interaction is probably caused by hydrogen bonding between S207(5.46), the TM5 bulge microswitch, and the ligand. In the TM domain, agonists induce a contraction between L284(6.46) and F321(7.48) (*Figure 5a and b*) compared to non-agonists. Both of these residues face the lipid bilayer and are only weakly interacting in the simulations with agonists bound, but are located in a hotspot region for activation. L284(6.46) is located just above the part of TM6 that kinks upon activation. The identified feature essentially connects the binding site and PIF motif to the NPxxY motif. F282(6.44) of the PIF motif is close to L284(6.46). T118(3.37), which was identified as important near the TM5 bulge (*Figure 5b*), is only one helix turn above I121(3.40) of the PIF motif in the connector region. Thus, the connector region is likely a driving factor behind the allosteric communication between the ligand and G protein-binding sites, which influences the region surrounding F321(7.48). F321(7.48) is located next to the NPxxY motif, which undergoes a twist upon activation, and is part of the important hotspot on TM7. In this region, our machine-learning analysis also identified that the backbone carbonyl of S319(7.46) formed a hydrogen bond with the side chain of N51(1.50) on TM1 for agonists, whereas this interaction was destabilized for the other ligands (*Figure 5d*). N51(1.50) is one of the most conserved residues across class A GPCRs (*Isberg et al., 2014*) and stabilizes a water network together with D79(2.50) and Y326(7.53) in the inactive receptor (*Cherezov et al., 2007*; *Venkatakrishnan et al., 2019*). Thus, this agonist-specific interaction, together with the D79(2.50)-N322(7.49) interaction (Fig. S7c and S8c), may promote dehydration of the water-filled cavity around conserved residue D79(2.50) and a twist of the NPxxY motif. Overall, agonists favored contractions between local inter-residue distances compared to non-agonists. By inspecting the most substantially changing large-scale distances, we also identified a contraction of the entire protein for agonist-bound receptors, as reflected by the decrease in distance between S203(5.43) and E338(8.56) on helix 8 (H8) and between S207(5.46) and V307(7.33) across the orthosteric binding site (*Figure 5a and c*).

Near the NPxxY motif, we found that the agonists stabilized different TM6 and TM7 orientations (*Figure 5d and e*). Adrenaline favored the most active-like NPxxY motif, which was also maintained

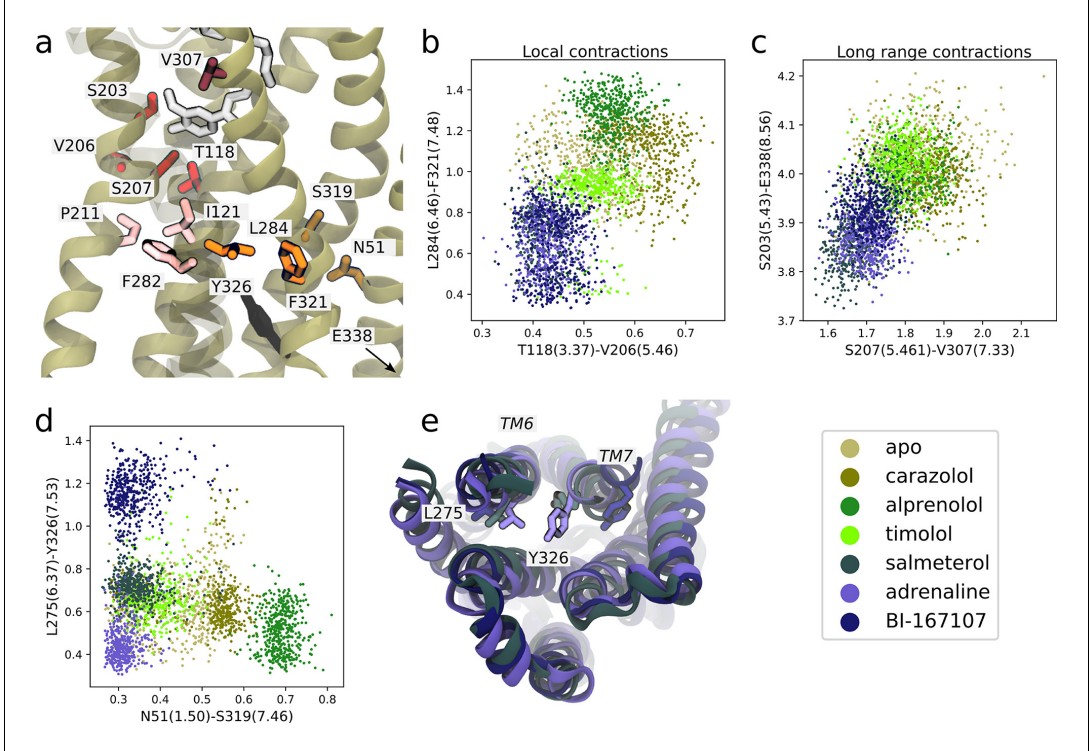

**Figure 5.** Molecular basis for agonists' control of receptor activation. (a) Molecular basis for agonists' control of receptor activation. S203(5.43) and S207 (5.46) (red sticks) are part of the transmembrane 5 (TM5) bulge, which forms direct contacts with the ligand. V206(5.46) forms van der Waals interactions with T118(3.37) (red), which is located above I121(3.40) of the PIF motif in the connector region (pink sticks). TM6 and TM7, highlighted as L284(6.46) (orange sticks) close to F282(6.44) of the PIF motif and F321(7.48) (orange sticks) above the NPxxY motif, move closer together in the presence of agonists. Half a helix turn above F321(7.48), S319(7.46) forms a backbone interaction with the side chain of N51(1.50) for agonist-bound receptors, whereas water molecules interact with these residues for non-agonists. Together with TM7-ligand contacts in the orthosteric site, these interaction pathways stabilize the second important hotspot on TM7 close to the NPxxY motif (Y326(7.53) shown in black). (b) The T118(3.37)-V206(5.46) distance near the orthosteric site against the L284(6.46)-F321(7.48) distance in the TM region. Agonists contract both of these regions. (c) The distance across the orthosteric site between S207(5.461) and V307(7.33) (dark red in [a]) against the S203(5.43)-E338(8.56) distance across the TM domain. Agonists stabilize more compact receptor conformations. (d) The N51(1.50)-S319(7.46) distance against the L275(6.37)-Y326(7.53) distance. Agonists share the common feature of stabilizing the N51(1.50)-S319(7.46) backbone interaction, but form different NPxxY orientations, shown as the distance from Y326(7.53) to L275(6.37). (e) The three agonists stabilize slightly different TM6 and TM7 orientations, here illustrated by the distance between L275(6.37) and Y326 (7.53). Adrenaline (purple) induces an active-like NPxxY motif, whereas BI-167107 (dark blue) stabilizes an inactive-like motif. The salmeterol-bound receptor (slate gray) adopts a distinct Y326(7.53) orientation.

The online version of this article includes the following figure supplement(s) for figure 5:

**Figure supplement 1.** Three distinct NPxxY conformations stabilized by salmeterol.

throughout its activation path (Fig. S7d), with Y326(7.53) closer to L275(6.37) (*Figure 5d*). Salmeterol stabilized a distinct NPxxY conformation, which was also observed in the activation path ensemble (*Figure 5—figure supplement 1*), in which Y326(7.53) underwent a rotation, bringing the tyrosine's side chain further into the interface between TM6 and TM7. This pose is reminiscent of conformations suggested by [19]F NMR studies on the $\beta_1$-adrenergic receptor (*Frei et al., 2020*). BI-167107 stabilized an inactive-like NPxxY motif with Y326(7.53) pointing away from L275(6.37) (*Figure 5d*).

In general, our data-driven approach automatically identified highly conserved residues involved in receptor activation (*Figure 4*). A noteworthy example is the D79(2.50) cavity, which is partially formed by strongly conserved residues S319(7.46), N51(1.50), and D79(2.50) and N322(7.49) (*Isberg et al., 2014*), and mutation of these may lead to non-functional receptors (*Chung et al., 1988*). Another example is the identification of V206(5.46), S207(5.46), and the PIF motif as a key region for allosteric communication. V206(5.46) and S207(5.46) were shown to interact with a recently discovered negative allosteric modulator that binds in an extrahelical site adjacent to the PIF motif (*Liu et al., 2020*). Our results do not only illustrate the usefulness of MD combined with

data-driven analysis; they allow us to identify potential allosteric sites that can be targeted by ligands, and reveal that, against our expectations, signaling hotspots near the NPxxY motif, far away from the orthosteric site, experience the largest ligand-induced conformational heterogeneity.

## Discussion

Following the progress in GPCR research, it has become evident that a simple two-state model of activation is an oversimplification with considerable limitations. To explain biased and partial agonism, there is a need for a more comprehensive model. Many ligands have been characterized as full, partial, or biased agonists (*van der Westhuizen et al., 2014*). However, a systematic characterization of the molecular mechanisms which transmit this allosteric communication across the cell membrane remains elusive. Researchers have successfully managed to discriminate between arrestin and G protein bias using spectroscopic probes (*Lamichhane et al., 2020*; *Liu et al., 2012*), but the conformational changes and dynamics of many microswitches cannot be captured in a single measurement. MD simulations have the potential to provide additional insights thanks to the atomistic-level description they enable (*Lamim Ribeiro and Filizola, 2019*). Whereas the free energy landscapes clearly show that ligands influence several microswitches, a direct comparison between free energy profiles may be misleading if the other orthogonal microswitches are ignored. For example, the local minima in the connector ΔRMSD landscapes are located at similar positions for all ligands, but our analysis clearly shows that the overall conformational ensembles differ (*Figure 2a* and *Figure 2—figure supplement 7b*). Such projections onto single variables can thus obscure major differences in other microswitches. To address this limitation, we have used data-driven analysis methods, which are better suited for handling high-dimensional data than mere visual inspection, and found that there were indeed significant conformational differences in the states stabilized by the different ligands. Remarkably, this protocol automatically identified both receptor-specific and conserved motifs considered to be of significance for GPCR activation, such as the TM5 bulge and the NPxxY motif, as important. The different dimensionality reduction techniques found similar partitioning of the data, which strongly indicates that the results are not due to fortuitous parameter tuning or method choice. One of the remaining enigmas in GPCR research is to understand how the same overall activation mechanism can be conserved in spite of the fact that very different ligands are recognized by the family. Our machine learning-inspired data analysis protocol provides an unbiased approach to identify key features of activation for different receptor types.

The derivation of free energy landscapes and the corresponding microswitch expectation values provide a tool for estimating the stability of activation states, and thus also the relative efficacy of different ligands. Given the high correlations between microswitch expectation values and experimental data, we anticipate that microswitches located in the orthosteric and connector regions can be used for future predictions of ligand efficacy. Additionally, an advantage of the methods used in this study is that results were derived from several simulation replicas, which reduces the statistical error related to the stochastic nature of MD simulations on short time scales. Since we allow the strings to diffuse around the converged equilibrium pathway for many iterations, the statistical error in the microswitch expectation values and energy landscapes is small, although it may be somewhat underestimated since swarm trajectories launched from the same point are correlated to each other. The systematic error is likely bigger for reasons related to the choice of force field (*Guvench and MacKerell, 2008*). An important limitation is that the string method can only identify one out of multiple activation pathways. Our control simulation starting from the inactive crystal structure, which converged to a different active state, indeed demonstrated this issue and highlights the importance of starting from a well-chosen input pathway.

Understanding ligand control of receptor activation is of great interest in drug development. The conformational selection of intracellular binding partners enables the construction of molecules or nanobodies that have a high binding affinity only in combination with a specific ligand (*Sencanski et al., 2019*), which has been utilized in structure determination (*Masureel et al., 2018*; *Rasmussen et al., 2011a*; *Ring et al., 2013*; *Staus et al., 2016*). Similarly, a well-designed nanobody or allosteric modulator could enhance or block the binding of a specific compound (*Staus et al., 2016*). However, in this study, the ligands were bound to the receptor throughout the simulations and no intracellular binding partner was considered. GPCRs only assume their fully active state in the presence of an intracellular binding partner (*Gregorio et al., 2017*; *Manglik et al., 2015*;

*Nygaard et al., 2013*). The β₂AR undergoes a basal activity, where it fluctuates between active-like and inactive states (*Gregorio et al., 2017*; *Lamichhane et al., 2015*), which can be inferred from the relatively low free energy difference between the basins in the energy landscapes. While extracellular and TM microswitches only require an agonist to maintain their active state conformations, intracellular microswitches interact with an intracellular binding partner directly and are more affected by its absence, which is likely why correlations between microswitch expectation value and Emax are worse for the intracellular microswitches.

Other important aspects of receptor–ligand interactions, such as identifying the binding pose of novel compounds or estimating a ligand's binding free energy, cannot be estimated with this approach. There are other complementary techniques such as free energy perturbation methods (*Cournia et al., 2017*; *Matricon et al., 2021*; *Matricon et al., 2017*), which can be used to rigorously estimate binding affinity. Combined with enhanced sampling MD, we move toward having a complete toolkit for in silico drug design for development of high-affinity GPCR ligands with a specific efficacy.

Despite the increasing number of β₂AR structures, the receptor has not been solved in a form bound to arrestin. However, conformations observed in our simulations share properties with other arrestin-bound states observed for other receptors. For example, the supervised learning methods identified the C-terminal and H8 below the NPxxY motif as relatively important (*Figure 4a*, *Figure 4—figure supplements 1–4*). This region is known to be stabilized in ligand-dependent states for the angiotensin II type 1 receptor (*Suomivuori et al., 2020*; *Wingler et al., 2019*). While the reorganization of H8 may be a secondary effect due to modulation of the NPxxY motif, this region could be important for arrestin recruitment (*Lally et al., 2017*; *Staus et al., 2018*). Salmeterol's distinct NPxxY state only formed in combination with a lost interaction between salmeterol and S207 (5.46) and S203(5.43), which is remarkable considering that the two binding sites are believed to be only loosely coupled (*Dror et al., 2011*; *Fleetwood et al., 2020b*). A similar phenomenon has been reported in a recent structure of β₁AR (*Lee et al., 2020*), where the corresponding serines in the orthosteric site experienced weakened interactions to the biased agonist formoterol. The fact that Y (7.53) forms contacts with arrestin for β₁AR (*Lee et al., 2020*) suggests that our derived β₂AR conformation may have biased signaling properties.

Our results show that the activation pathways as well as the stabilized states are significantly altered upon ligand binding, and that ligands with shared efficacy profiles generate similar, albeit not identical, ensembles of states. It therefore cannot be taken for granted that two ligands which lead to a similar downstream response necessarily stabilize the same receptor conformations. As we considered several compounds in this study, similarities and differences between different compound classes emerged. The results in this study provide a good starting point for further analysis and allowed us to catch a glimpse of the complexity underlying receptor signaling. A thorough quantification of biased and partial agonism will require studies of ligands that stimulate various signaling pathways to different extents.

## Conclusion

In this study, we derived the activation free energy of the β₂AR bound to ligands with different efficacy profiles using enhanced sampling MD simulations. We found a strong correlation between cellular response and the computed expectation values of the upper and TM microswitches, which suggests that our approach holds predictive power. Not only did the results show how ligands control the population of states, they also modulate the conformational ensemble of states by tuning important allosteric microswitches in the vicinity of the G protein-binding site. By inspecting the contacts formed for agonists and non-agonist ligands, we identified an allosteric pathway between the two binding sites and a large heterogeneity of TM7 states. Our results show how enhanced sampling MD simulations of GPCRs bound to ligands with various activation profiles, in combination with a data-driven analysis, provide the means for generating a comprehensive view of the complex signaling landscape of GPCRs. We anticipate that our protocol can be used together with other computational methods to understand GPCR signaling at the molecular level and provide insights that make it possible to design ligands with specific efficacy profiles.

## Materials and methods

### Simulation system configuration

We initiated simulation systems from a nanobody-bound active-state BI-167017-bound structure (PDB ID: 3P0G) (*Rasmussen et al., 2011a*) and an inactive carazolol-bound structure (PDB ID: 2RH1) (*Cherezov et al., 2007*) in CHARMM-GUI (*Lee et al., 2016*) with the CHARMM36m force field (*Huang et al., 2017*). Since the two structures are missing certain residues and have different thermostabilizing mutations, we used GPCRDB's (*Isberg et al., 2014*) improved version of 2RH1, removed residues not present in 3P0G, and mutated E27(1.26) to Q, a residue frequently found in the human population (*Dallongeville et al., 2003*). As a result, the two simulation systems were identical. Following the protocol of a previous study (*Fleetwood et al., 2020b*), we reversed the N187E in the crystallized structures, protonated E122(3.41), and protonated the two histidines H172 (4.64) and H178(4.70) at their epsilon positions. The receptor was embedded in a POPC membrane bilayer (*Klauda et al., 2010*) of 180 molecules, then solvated in a 0.15M concentration of neutralizing sodium and chloride ions with 79 TIP3P water molecules (*Jorgensen et al., 1983*) per lipid molecule. We performed the MD simulations with GROMACS 2018.6 (*Abraham et al., 2015*) patched with PLUMED (*Tribello et al., 2014*). Ligands present in PDB structures 2RH1 (*Cherezov et al., 2007*), 3NYA (*Wacker et al., 2010*), 3D4S (*Hanson et al., 2008*), 6MXT (*Masureel et al., 2018*), and 4LDO (*Ring et al., 2013*) were inserted into the 3P0G structure after alignment of residues interacting with the ligand. Input files required to run the simulations in this study are available online (*Fleetwood, 2019a*).

### String method with swarms of trajectories

We used an optimized version of the string method with swarms of trajectories to enhance sampling and to estimate the free energy along various microswitches (*Fleetwood, 2020a*; *Pan et al., 2008*; *Lev et al., 2017*). This method finds the most probable transition pathway between two end states in a high-dimensional space spanned by a set of CVs (in this context synonymous with *reaction coordinates*). Given the initial guess of *points* distributed along a *string* in CV space, in this case the transition path from the previously published apo simulation, the 3P0G-initiated systems were equilibrated by running 200 ns steered MD simulations along the string with force constant per CV of 3366 kJ/mol*nm$^2$ scaled by its estimated importance (more details in the following section). This was then followed by a 7 ns initial restrained equilibration at every point. Next, a *swarm* of 10-ps-long trajectories were launched from the output coordinates of every restrained simulation. The average drift of the swarm was computed as the mean displacement of the short swarm trajectories in CV space, which is proportional to the gradient of the free energy landscape. Every swarm consisted of 16–32 trajectories and the exact number was determined adaptively to converge the drift vector. The points were displaced according to their drift and re-aligned along the string to maximize the number of transitions between neighboring points. The string was updated iteratively with 30 ps of restrained equilibration per point, followed by a batch of swarms and string reparametrization. Gradually, the string relaxed into the most probable transition path connecting energetically stable intermediates between the two endpoints. We ran the simulations for 305 iterations, requiring an aggregated simulation time of 4.3 μs per ligand.

As a control, we also initiated a string from the carazolol-bound inactive state structure 2RH1. Steered MD from the inactive to the active state resulted in a slight unfolding of the intracellular part of TM6. Instead, we followed a slightly different protocol (*Lev et al., 2017*) and initiated the pathway by applying 200 ns targeted MD with a stronger 100 MJ/mol*nm$^2$ force constant on all protein-heavy atoms. From this pathway, the string with swarms of trajectories was launched using the same CVs and algorithm as described above.

Convergence is generally reached when the string diffuses around an equilibrium position. Due to MD simulations' stochasticity, two strings from subsequent iterations may therefore differ even when the system has reached equilibrium. To evaluate the convergence we averaged the strings over 60 iterations, and stopped sampling every simulation after 305 iterations, at which point the average strings for all ligand–receptor complexes had converged.

## Kinetically trapped active-like state sampling

In order to quantitatively sample the most stabilized state kinetically accessible from the starting structure without applying an artificial force on the system, we developed an adaptive sampling protocol. A single swarm with twenty-four 7.5-ns-long trajectories was launched from the same initial active configuration as described in the previous paragraph. The swarm's center point, $c$, in CV space was taken as the mean of the trajectories' endpoint coordinates, $x_i$. Next, we computed the average distance, $d$, from the endpoints to the center and assigned every replica, $i$, a weight, $w_i(x)$ =exp(-($|x_i-c|$ / $d$) 2). New trajectories were iteratively seeded by extending $n_i \propto w_i/\sum_j w_j$ copies of each replica, keeping the total number of trajectories fixed to 24. With this approach, only replicas close to the center were extended and the ensemble of trajectories eventually diffused around a single equilibrated state.

We evaluated convergence by monitoring the distance between the center point of subsequent iterations until it converged to a constant value, which occurred within eight iterations for all systems. To demonstrate the robustness of the results, we split the walkers into three sub-groups for cross validation analysis.

## Collective variable selection

We derived the CVs for the string method with swarms of trajectories in a data-driven manner from the swarm coordinates of the apo simulation's final iteration using demystifying (*Fleetwood et al., 2020c*), a software which utilizes machine learning tools and dimensionality reduction methods to identify important features from MD simulation trajectories. As features, we chose inverse inter-residue C-α distances and filtered them to only include those which sampled values in the interval 6–8 Å. We then used the features to train a restricted Boltzmann machine (RBM) (*Smolensky, 1986*). An RBM is a single-layer neural network with a number of hidden components (two in this manuscript), equivalent to a fully connected bipartite graph. Upon training, the network is tuned to fit a certain statistical model, which maximizes the joint probability between the components in the input layer and the hidden layer (*Pedregosa et al., 2011*). The input features were ranked by their importance using layer-wise relevance propagation, an algorithm originally developed to identify important pixels in image classification problems (*Montavon et al., 2018*). Since we used stochastic solvers, the results were averaged over the results from 50 independent RBMs. Only CVs with an estimated importance above 0.33 were included in the final set (*Table 2*). Every CV was first scaled unitless in order to keep all values between 0 and 1, then rescaled according to its importance, so that the restraining force and the drift in the swarms of trajectories method would better emphasize the conformational changes along important degrees of freedom. Finally, we derived a new pathway in the resulting CV space by interpolating between the restrained points of the converged apo simulation. All string simulations used these CVs and this new pathway as a starting point to launch swarms except for the previously published apo and BI-167107 systems.

## Free energy estimation

Free energy landscapes were estimated during post-processing by discretizing a grid along a chosen set of variables and constructing a regularized transition matrix from the swarm trajectories' transitions between bins. We then derived the resulting free energy landscape from the stationary probability distribution of the transition matrix using Boltzmann inversion (*Fleetwood et al., 2020b*; *Flood et al., 2019*; *Lev et al., 2017*). To estimate the convergence of the free energy landscapes, we applied a Bayesian Markov chain Monte Carlo method (*Harrigan et al., 2017*) to sample 1000 different transition matrices from the dataset, each with a corresponding probability distribution and free energy landscape. From these we could estimate standard statistical properties such as the mean value and sample standard error. Swarm trajectories from a well-sampled equilibrium ensemble with multiple transitions between states will generate a narrower distribution of free energy values than trajectories drawn from a non-equilibrium process or a poorly sampled system.

## Microswitch expectation values and equilibrium between states

To quantify the effect of a ligand on different microswitches, we computed the expectation value and the relative difference in free energy between the active and inactive state (ΔG) of individual microswitches. ΔG was obtained by integrating the free energy landscape over the active and

inactive basins, respectively. We defined approximate state boundaries by visual inspection of the free energy landscapes and the crystal structures.

We then evaluated the correlation of these two values with experimental measurements of cellular response to ligand binding (*van der Westhuizen et al., 2014*) using linear regression in the python software package SciPy (Enthought Inc, Austin, TX; *Millman and Aivazis, 2011*). Finally, we applied the derived linear relationship to predict the efficacy of the two ligands not part of the experimental dataset, BI-167107 and carazolol, based on their microswitch expectation values.

## Supervised and unsupervised feature extraction and learning

We analyzed the trajectories from the last iteration of the adaptive state sampling protocol and the swarms of trajectories method with various dimensionality reduction methods. To identify conformational differences induced by the ligands, we performed supervised and unsupervised feature extraction with the software demystifying (*Fleetwood et al., 2020c*), and projected important features onto snake plot templates downloaded from GPCRDB (*Isberg et al., 2014*). As features, we used scaled inverse closest-heavy atom distances. Furthermore, we performed unsupervised dimensionality reduction in Scikit-learn (*Pedregosa et al., 2011*) with PCA (*Tipping and Bishop, 1999*), MDS (*Borg and Groenen, 2005*) with a Euclidean distance metric and t-SNE (*van der Maaten and Hinton, 2008* ), and projected the simulation snapshots onto the reduced feature spaces. We constructed a similarity metric by taking the average Euclidian distance between all simulation snapshots in two classes, and normalized the class similarities between 0 and 1, with higher values representing similar classes.

Moreover, we computed how important individual residues were for discriminating between agonists and non-agonists and to distinguish all ligands from each other, using a symmetrized version of the KL divergence (*Fleetwood et al., 2020c*; *Kullback and Leibler, 1951*). With this approach, two residues constituting a distance were scored as important if the active-like states formed non-overlapping distance distributions, corresponding to a high KL divergence. As a control, we evaluated the important features learned by a RF classifier (*Ho, 1995*), a machine learning model constructed by an ensemble of randomly instantiated decision trees. The importance of inter-residue distances was computed during training by normalizing the RF's mean decrease impurity (*Breiman et al., 1984*; *Pedregosa et al., 2011*), which measures how frequently a distance is used to split the decision trees.

Unsupervised feature extraction was performed with PCA (*Tipping and Bishop, 1999*), which transformed the input dataset of distances, $\mathbf{F}$, into to a set of orthogonal variables called PCs. The PCs are equivalent to the eigenvectors of $\mathbf{F^T F}$, and their eigenvalues measure how much of the variance in $\mathbf{F}$ they cover. Thus, by multiplying the PCs with their egeinvalues, and projecting them back onto the input features, we obtained an estimate of importance corresponding to how much the individual distances contributed to the variance in $\mathbf{F}$.

Software code to reproduce the results in this manuscript is available online (*Fleetwood, 2020a*; *Fleetwood, 2019a*; *Fleetwood, 2019b*).

## Acknowledgements

This work was supported by grants from the Gustafsson Foundation, Knut and Alice Wallenberg foundation (2019.0130), and Science for Life Laboratory to JC and LD. The work was also supported by grants from the Swedish Research Council (2017–04676) and the Swedish strategic research program eSSENCE to JC. The simulations were performed on resources provided by the Swedish National Infrastructure for Computing (SNIC) at PDC Centre for High Performance Computing (PDC-HPC).

## Additional information

### Competing interests

Lucie Delemotte: Reviewing editor, *eLife*. The other authors declare that no competing interests exist.

## Funding

| Funder | Grant reference number | Author |
|---|---|---|
| Göran Gustafssons Stiftelse för Naturvetenskaplig och Medicinsk Forskning | | Jens Carlsson Lucie Delemotte |
| Science for Life Laboratory | | Jens Carlsson Lucie Delemotte |
| Vetenskapsrådet | 2017-4676 | Jens Carlsson |
| Swedish strategic research program eSSENCE | | Jens Carlsson |
| Knut och Alice Wallenbergs Stiftelse | | Jens Carlsson Lucie Delemotte |

The funders had no role in study design, data collection and interpretation, or the decision to submit the work for publication.

## Author contributions

Oliver Fleetwood, Conceptualization, Software, Formal analysis, Validation, Investigation, Visualization, Methodology, Writing - original draft, Writing - review and editing; Jens Carlsson, Conceptualization, Supervision, Funding acquisition, Validation, Investigation, Visualization, Writing - review and editing; Lucie Delemotte, Conceptualization, Formal analysis, Supervision, Funding acquisition, Validation, Investigation, Visualization, Writing - review and editing

## Author ORCIDs

Oliver Fleetwood (iD) https://orcid.org/0000-0002-4277-2661
Jens Carlsson (iD) https://orcid.org/0000-0003-4623-2977
Lucie Delemotte (iD) https://orcid.org/0000-0002-0828-3899

## Decision letter and Author response

Decision letter https://doi.org/10.7554/eLife.60715.sa1
Author response https://doi.org/10.7554/eLife.60715.sa2

# Additional files

## Supplementary files

• Transparent reporting form

## Data availability

The data necessary to reproduce the findings presented in this paper can be found on OSF (DOI 10.17605/OSF.IO/6XPYV). The code used to run and analyze simulations has been deposited on GitHub (https://github.com/delemottelab/demystifying, https://github.com/delemottelab/gpcr-string-method-2019 and https://github.com/delemottelab/state-sampling; copies archived at https://archive.softwareheritage.org/swh:1:rev:e8527b52d5fbe0570cd391921ecda5aefceb797a/, https://archive.softwareheritage.org/swh:1:rev:bc3b7ce2e74e5ac95644d57a1b24f717a7ec74a4/, https://archive.softwareheritage.org/swh:1:rev:f0f56430ce581f0338771c126da212ecc2f218a0/).

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
