## [Decision Letter]

**Acceptance summary:**

This manuscript reports advanced molecular dynamics simulations to describe

β_2_ adrenergic receptor modulation, expanding significantly on existing knowledge. The study has made use of an atomistic string method to measure the effects of agonists, antagonists and inverse agonists and to understand how ligands affect GPCR activity. The authors have presented sufficient analysis to demonstrate statistical significance of the data and have made connections to experimental measurements that are well described in the revised manuscript. Overall, this is a high-level computational study of biological significance that will be impressive to many *eLife* readers.

**Decision letter after peer review:**

Thank you for submitting your article "Identification of ligand-specific G protein-coupled receptor states and prediction of efficacy via data-driven modeling" for consideration by *eLife*. Your article has been reviewed by three peer reviewers, one of whom is a member of our Board of Reviewing Editors, and the evaluation has been overseen by José Faraldo-Gómez as the Senior Editor. The reviewers have opted to remain anonymous.

Based on the reviews received, which are enclosed below, as well as a follow-up discussion amongst reviewers and editors, we regret to inform you that we cannot publish your manuscript in its current form. However, the reviewers have agreed to re-evaluate a revised version of the manuscript that convincingly addresses the concerns raised. While we encourage you to submit a revision, please note that a positive outcome is not guaranteed.

The reviewers have appreciated many aspects of the string method solution for the allosteric modulation of the β_2_ adrenergic receptor, but were concerned about the reliability of the simulations, their connections to experiments and the absence of discussion on the relationship to past GPCR activation studies. Concern is expressed about the use of Emax as the sole proxy for activated conformation, as well as a lack of connection of free energy calculations to experiments. A critical point to address is the absence of a description of the statistical reliability and error estimates from the string method solutions, as well as the dependence of the converged solutions on the starting conformation. This necessitates at least one demonstration that reliable results emerge with a very different initial structure. A significantly revised manuscript would need to adequately address the above concerns to the satisfaction of the reviewers.

Essential revisions:

The full reviews have been included below. Essential revisions include:

* Proof of independence of string solutions on starting structure.

* Proof of statistical reliability and error estimates.

* Improved connections to experiments that do not rely solely on Emax values.

* Improved relationship to past studies.

*Reviewer #1:*

This manuscript reports advanced MD simulation to describe the allosteric modulation of the β_2_ adrenergic receptor, expanding on the authors' recent simulations (Biochemistry 2020, 59:880-891) that explored communication of ligand binding to the G protein-binding site via "microswitches" within the protein. This study uses the same string method to examine the effects of full and biased agonists, antagonists and inverse agonists using previously solved Xray structures. The study also calls on a series of data-based analysis methods to seek answers to the question of how different ligands communicate their changes and affect GPCR activity. Observation of stabilization of active states for agonists and inactive states for other ligands is an important achievement for MD simulation, even if reproducing known experimental results. Also, characterizing how the distribution of kinetically stabilized (active-like) states are controlled by ligands, and correlations between microswitch expectation values and cAMP response experiments are important. However, the manuscript is highly technical and jargon-based such that many readers will not follow the analysis. The statistical reliability and dependence of the converged string solutions on starting conformation do not appear to have been tested. Finally, relationship between this and the previous manuscript (Fleetwood et al., 2020), as well as how results distinguish themselves from knowledge in the field could be made more clear.

String methods are notorious for depending on starting structure as well as the number and choice of collective variables, and may converge slowly on the optimal path, if at all. The authors do not show plots of convergence (how each PMF changes over #iterations), and do not appear to undertake tests for reproducibility. i.e. Starting from very different initial structures to show that the same final pathways emerge, leading to consistent free energy profiles. For example, starting again with an initial structure for an inverse agonist instead of one for an agonist and show the same results are computed. I note that in the Discussion it is said that several simulation replicas were used to reduce error. But to what does this apply (apparently not to the swarms)? Also, collective variables (CVs) are those defined in the previous study (Fleetwood et al., 2020) but what if the set of CVs is changed or the number of CVs is reduced/increased? While I do not suggest redoing with a different set, how does the reader judge the robustness of the CV-reliant results?

The method for free energy calculation uses a transition matrix on a CV grid with stationary solution. In the subsection “Free energy estimation”, the authors cite Pan, Sezer and Roux, 2008 (and Fleetwood et al., 2020), but it is not clear that Pan, Sezer and Roux, 2008, did this, and in fact I think the first to do this from a swarms of trajectories solution was Lev et al., 2017 (see also Flood, et al., 2019), not cited here.

Having computed free energy projections along collective variables, receptor stability in relation to efficacy instead relies on estimation values for variables and not free energies, and the justifications for this could be improved. The authors write that the width and depth of the basins determine the most probable state projected into that space. This sentence is not clear, but its meaning is important, because ultimately the relative values of free energies are discussed. If the 2D maps are valid projections of the full configurational space (albeit with sampling guided/biased by the string), then Boltzmann integrals over each basin should yield a valid equilibrium constant. I presume the concern is that projections along different pairs of CVs can lead to different apparent free energies, because different CVs map out different proportions of phase space, and potentially envelop multiple states of the system, but does this mean a Boltzmann integral over a site is not a true thermodynamic quantity? In the Discussion, the authors return to add that examination of one projection can overlook what is happening in other coordinates/switches, but more discussion/justification is needed. Related, the authors state that they have "accurately captured the relative stability of states", but this is confusing given the relative stability of states from the maps appears to have been discredited.

Instead of using free energies, the authors correlate "expectation values" of their CVs to experiments in the bottom of Figure 2. I assume a plot against free energy was abandoned as it was not working as planned? Ideally one would want to see free energies plotted against experimental efficacy, because estimation values of variables such as TM5…, may correlate to efficacy, but not uniquely map to shifting state equilibrium.

Many analysis tasks were completed by existing packages for which the meanings are not obvious. e.g. Demystifying, Scikit-learn… Materials and methods, even if accepted, deserve a sentence to explain and motivate. e.g. CVs were short inverse inter-residue distances used to train a restructured Boltzmann machine (the principles of this machine could be explained simply). The subsection “Data driven analysis reveals that ligands stabilize unique states” is highly technical and not well explained. While many will know about PCA, less will be known about MDS and T-SNE. The ability to identify signaling hotspots in Figure 4 is impressive, leading to a model for allosteric communication. But the machine learning approaches used come across as black box and require better physical interpretation.

*Reviewer #2:*

This manuscript describes the use of enhanced sampling molecular dynamics to calculate free energy landscapes for the β_2_ adrenergic receptor. A particular focus is the conformational dynamics and thermodynamics of microswitches that change conformation upon receptor activation. A variety of ligands are investigated to identify shared and divergent features of receptor conformational modulation. Unbiased methods are shown to identify known conformational switches as key regulators of receptor activation.

Overall the manuscript is interesting and clearly written. Figures are very clear and well presented. The subject matter has been very extensively studied in the GPCR family as a whole and in the β_2_ receptor in particular, and the results presented here largely align with existing understanding of GPCR activation and conformational regulation by ligands. A major concern is the question of whether the results presented here truly enhance our understanding of GPCR activation, or simply confirm things that are already known. Many groups have published detailed analysis of similar questions to those presented here, including the Dror, Nagarajan, and McCammon groups, among others. There is very little discussion of this prior work, which makes it difficult to see how the present manuscript fits within the broader context of GPCR activation molecular dynamics analysis.

A significant technical concern is the reliance on cAMP Emax values as the sole experimental validation. A prospective experimental test of hypotheses generated here would significantly enhance the manuscript. Barring this, a more extensive comparison of computational results with experimental data is essential in my view. The cAMP pathway is highly amplified, which may confound analysis linking Emax values directly to conformational equilibria. Manglik et al., 2015, presents an actual biophysical analysis of conformational equilibria (albeit with fewer ligands) and comparison to this would be helpful. Do relative energy predictions match these experimental observations?

A smaller issue is that the utility of the results presented here appears to be a bit overstated. For example, it is claimed (Introduction) that the results provide insight into how ligands with specific efficacy profiles can be designed. To support this statement, the authors should present examples of compounds they have designed based on their results, together with experimental data confirming these molecules have the intended efficacy profiles.

*Reviewer #3:*

This is an interesting paper, and I like to attempt to connect analysis of allostery to function. However, I'm extremely concerned about statistical uncertainty - it's not really discussed, and it would be easy to chalk all of the results up to limited sampling. It will be important for the authors to demonstrate this isn't the case.

Basically, my concern is that there's essentially no mention of statistical uncertainty or convergence anywhere in the document. One of the major claims is that the different agonists each populate a distinct substate - this is an incredibly important and interesting observation if true, but could also be explained by saying each simulation wandered in its own space and didn't have time to explore anywhere else. If it were run again, it might wander into a totally different place. I don't have a great feel for how rapidly swarms explore, but I do know the total sampling time of 1.4 µs per ligand sounds awfully small. There are examples in the literature showing that conventional simulations several times this length are not converged as far the configurations in the ligand binding pocket go (for example, Leioatts et al., Biophysical Journal, 2015, or some of the work from Dror's group).

I have a couple of ideas for how the authors could convince me this isn't just a sampling artifact. The best would probably be to pick one ligand and redo the whole calculation 4 more times, start to finish - looking at the variation between those replicates would be a decent estimate of the uncertainty. Ideally, they'd do this for all of the ligands, but I recognize that's not a reasonable request, which is why I suggest doing it for 1 ligand.

A weaker test would be to break the individual ligand calculations into blocks, somehow - I'm not totally sure how to do it - and show that the blocks are self-similar. If they're all wandering through the whole of the blob in Figure 3, you're more likely to be ok. If each block populates a discrete chunk of the blob (and the overall swarm data doesn't revisit), then there's a big problem.

I'd want to see error estimates on the free energies in Figure 2, plus a redo of the dimension reduction in Figure 3 to see if the ligands still separate more than the replicates of 1 ligand.

On a more minor note, the use of p-values in the subsection “Ligands control efficacy by reshaping microswitches’ probability distributions”, isn't really correct. The point the authors are trying to make is that the computed value doesn't predict the experiment (for good reason), and the correlation coefficients make that point for you.

---

## [Author Response]

*Essential revisions:*

*The full reviews have been included below. Essential revisions include:*

** Proof of independence of string solutions on starting structure.*

** Proof of statistical reliability and error estimates.*

** Improved connections to experiments that do not rely solely on Emax values.*

** Improved relationship to past studies.*

1. Proof of statistical reliability and error estimates.

We have performed a number of steps to better demonstrate the statistical reliability of our results:

String simulations

The free energy estimates constitute the most important result derived from our string simulations (Fig.2). To better assess their convergence, we analyzed the convergence of the underlying transition matrix, T, of the swarm trajectories’ transitions between microstates. Since many factors go into constructing T, the impact of all the statistical uncertainties on the final free energy landscape cannot be directly derived. Instead, we applied a Bayesian approach and used Metropolis Markov chain Monte Carlo (MCMC) to sample over the posterior distribution of transition matrices (Harrigan et al., 2017), and thereby obtained a distribution of free energy profiles. From these we could compute the mean, standard deviation, whiskers and outliers etc., which are shown in Figure 2–figure supplement 7A and B. Although this convergence assessment was already included in the original manuscript, we acknowledge that we did not properly inform the reader of this analysis. This has been corrected in this revised manuscript.

As shown in Figure 2–figure supplement 7A and B, the error bars are typically small near the basins, and thus the standard error of the microswitch expectation values will also be small. The error bars may therefore be difficult to discern for some values in the correlation figures in Fig. 2. This use of bootstrapping analysis methods to assess convergence is common practice in the analysis of Markov State Modelling (Prinz et al., 2011). Note that if we have many transitions across free energy barriers, this approach is prone to underestimate the total error since systematic errors due to force field are not taken into account.We also monitored the convergence of the pathway for every simulation in CV space. Author response image 1 shows that after approximately 100 iterations the string does not move on average. Instead, it fluctuates around an equilibrium value.

**Author response image 1. respfig1:** Example of string pathway convergence plot with the progress of inactivation plotted against the highest ranked CV. Everystring has been averaged over 61 iterations. More details, including error bars, can be found in SI figures S1-S6

State sampling simulations

The state sampling simulations form the basis for Fig. 3-5. It is an iterative sampling approach where we track the center point in CV space of multiple walkers. By iteratively terminating walkers which drift far away from the center (since they may be considered as statistical outliers), we monitor how the center point converges to a fixed value. Figure 3 – figure supplement 3, which shows the distance between center points, indicates convergence for all systems within 8 iterations. This figure has now been updated to also show the standard error of the walkers’ distance to the center point. The distance between center points is smaller than the error and the walkers’ center is fluctuating around their equilibrium position. The distribution of walkers around this position is what we refer to as the pre-active state.

Following Reviewer #3’s suggestion, we also performed cross validation with a subset of the walkers and showed that the results are self-similar. In other words, by only considering one third of the walkers at the time, we arrived at very similar dimensionality reduction figures. We thus found that our claims regarding the similarities and differences between agonists and nonagonists still held after these convergence assessments. These results have been included in the revised Figure 3 – figure supplement 3.

Investigation of independence of string solutions on starting structure.

Our original starting method calculations were initiated from the active structure with PDB code 3P0G. To address the editor and reviewers’ comment, we initiated a new string simulation from the inactive state carazolol-bound structure with PDB code 2RH1. We note however that a fully active conformation is observed in the presence of intracellular binding (*e.g.* G protein), which we do not have in our system and that thus driving the system from the inactivated state to the activated one is expected to be energetically unfavorable and thus more challenging than the reverse. Apart from the difference in conformation, the two receptor structures were identical in terms of side chain protonation and treatment of loops and termini. The initial pathway was different too, as we applied targeted MD on the 2RH1-initiated structure toward the active structure, using the same CVs as before. After convergence, we obtained a similar pathway: calculating the expectation values of the microswitches for the ensemble derived from the system initiated from the inactive state structure, preserves the overall trend for carazolol to cluster away from agonists for transmembrane microswitches (Figure 2 – figures supplement 10).

Interestingly, the pathway is not entirely identical (Figure 2 – figures supplement 9) and some states in the activated region of the landscape, in particular, appear different. The states in the inactive region, on the other hand are equivalent. Since the fully activated state is only stabilized in the presence of an intracellular binding partner, we expect the activated state region to not be sampled fully adequately starting from the inactive state. We would thus recommend initiating simulations from active structures if such structures are available. In addition to this new simulation, we have previously shown that the swarm trajectories near the active state basin were similar to the G protein bound structure with PDB code 3SN6 (Fig. S3 in (Fleetwood et al., 2020), even more so than to the nanobody-bound structure 3P0G, which was used to initialize the system. This suggests that our results are not heavily dependent on the choice of starting structure when it is appropriately chosen.

2. Improved connections to experiments that do not rely solely on Emax values.

Our ambition was to present a quantitative comparison of microswitch simulations to experimental results. Quantitative comparison is notoriously difficult since it typically involves a comparison of physical properties across widely different length- and time-scales. In this study, we thus considered it an achievement that were able to correlate predicted expectation values for microswitches to experimental measurements of downstream signaling. Nevertheless, following the reviewers’ comments, we have complement these findings with a comparison to other types of experiments, especially to biophysical studies which, combined with our results, can provide mechanistic insights.

In the first part of the results section we make a qualitative comparison between the free energy landscapes for the apo, carazolol-bound, and BI-167107-bound receptor to distance distributions in spectroscopy experiments (Manglik et al., 2015). Since the “ionic lock” microswitch is frequently discussed in that and other papers, we have replaced the TM6-TM3 distance in Fig.2 and Figure 2 – figure supplement 9, 11, with a measurement that better reflects the ‘ionic lock distance’. This does not influence the overall results of the paper—the only difference is that we compute the distance from R131 to E268 instead of L266—but it may simplify the interpretation of the results for readers well acquainted with the GPCR literature. We also investigate the change in hydration of TM6’s intracellular part, which could explain an observed change in probe environment.

We also discuss the heterogeneity of states identified by these experiments, the existence of preactive states along the activation pathway and our single equilibrated states. Digging further into the molecular basis for the agonist-specific states, we identify the transmembrane region around the PIF motif to be an allosteric switch, which explains why this region it is a target of allosteric modulators (Liu et al., 2020) (see the final paragraph of the last result section). We also identify signalling hotspot around the NPxxY motif to be the major contributor to the heterogeneity of states, in agreement with spectroscopy studies (Frei et al., 2020; Lamichhane et al., 2020; Liu et al., 2012) (see also second to last result section). Finally, we have elaborated on the general result that our analysis methods tend to detect highly conserved residues in the transmembrane region, which experiments have identified to be critical for the receptor to function (see last result section). All new text is marked in blue, and we hope the reviewers will find it easy to locate these new experimental comparisons in the revised manuscript.

3. Improved relationship to past studies.

The revised manuscript includes more citations of past studies and describes them in greater detail. The introduction has been extended to include a new paragraph that describes previous state-of-the-art MD simultion studies of the β2AR. We also present methodological development from our previous paper (Fleetwood et al., 2020), while introducing changes that make this new manuscript understandable for a reader unfamiliar with our previous work. As described in the previous section, we also put our results into the context of previous experimental results throughout the text. Note however that given the number of studies of this receptor carried out so far, a thorough review of all previous studies is not possible in an original research article.

Regarding the novelty of our work:

One of the reviewer also questioned the novelty of our work. Although GPCR activation has been previously studied with MD simulations and that our work clearly builds on previous efforts, we believe that the results presented here enhance substabtially our understanding of GPCR activation and conformational regulation by ligands. This is made possible by the availability of high quality structures of β2AR which inform our computational work. We also note that this work is also enabled by the sampling procedures we have proposed, which allow the characterization of the conformational activation ensemble at a reasonable computational cost, and thus the repetition of the procedure for a set of ligands.

First, our study shows how intracellular and transmembrane microswitches *quantitatively* correlate with experimental downstream response. Second, we find that all ligands stabilize distinct pre-active states, but that ligands with similar pharmacological properties stabilize similar states. Although several results agree with previous experimental or computational findings, we provide novel detailed molecular insight into the mechanism and energetics of these states, thus enhancing understanding of GPCR activation. Finally, the manuscript takes a methodological leap toward quantitative data-driven analysis of datasets with comparatively many ligands, and thereby sets a new state-of-the-art for GPCR molecular dynamics analysis.

Reviewer #1:This manuscript reports advanced MD simulation to describe the allosteric modulation of the β_2_ adrenergic receptor, expanding on the authors' recent simulations (Biochemistry 2020, 59:880-891) that explored communication of ligand binding to the G-protein binding site via "microswitches" within the protein. This study uses the same string method to examine the effects of full and biased agonists, antagonists and inverse agonists using previously solved Xray structures. The study also calls on a series of data-based analysis methods to seek answers to the question of how different ligands communicate their changes and affect GPCR activity. Observation of stabilization of active states for agonists and inactive states for other ligands is an important achievement for MD simulation, even if reproducing known experimental results. Also, characterizing how the distribution of kinetically stabilized (active-like) states are controlled by ligands, and correlations between microswitch expectation values and cAMP response experiments are important. However, the manuscript is highly technical and jargon-based such that many readers will not follow the analysis. The statistical reliability and dependence of the converged string solutions on starting conformation do not appear to have been tested. Finally, relationship between this and the previous manuscript (Fleetwood et al., 2020), as well as how results distinguish themselves from knowledge in the field could be made more clear.

We thank the reviewer for this assessment. Regarding the high technical level, we have tried to make simplifications and clarifications throughout the revised manuscript. For example, we now give more background on what the software and machine learning tools do. We have also tried to better define the connection between this manuscript and the previous work. We hope the changes appear satisfactory.

String methods are notorious for depending on starting structure as well as the number and choice of collective variables, and may converge slowly on the optimal path, if at all. The authors do not show plots of convergence (how each PMF changes over #iterations), and do not appear to undertake tests for reproducibility. i.e. Starting from very different initial structures to show that the same final pathways emerge, leading to consistent free energy profiles. For example, starting again with an initial structure for an inverse agonist instead of one for an agonist and show the same results are computed. I note that in the Discussion it is said that several simulation replicas were used to reduce error. But to what does this apply (apparently not to the swarms)? Also, collective variables (CVs) are those defined in the previous study (Fleetwood et al., 2020), but what if the set of CVs is changed or the number of CVs is reduced/increased? While I do not suggest redoing with a different set, how does the reader judge the robustness of the CV-reliant results?

In addition to the arguments listed in the cover letter, we point out that the previously published apo and BI-167107-bound string simulations were performed with a different set of Collective Variables (CVs) and initiated from different pathways (Fleetwood et al., 2020) than the new ligand-receptor simulations presented in this manuscript. Despite the differences in simulation set-up, it is evident from Figure 2 that the free energy landscapes for the agonists adrenaline and salmeterol are more similar to BI-167107 than to the other ligands. The same holds for the apo free energy landscape, which is strikingly similar to the inverse agonists and antagonists. If the starting conditions and CVs alone defined the outcome, this would not be the case.

Regarding the convergence of the PMF, we previously performed cross validation over the string iterations for the TM5 bulge microswitch expectation values (Supplementary Table 2 in the original manuscript). The microswitch expectation values were consistent between the first and second half of the last 224 string iterations, which indicates that these values have converged and were sampled from an equilibrium process. Additionally, samples drawn from an unconverged distribution, would lead to a large standard deviation and/or statistical outliers in the free energy landscapes. Thus our statistical error analysis and convergence estimation provide much strong evidence of convergence.

The method for free energy calculation uses a transition matrix on a CV grid with stationary solution. In the subsection “Free energy estimation”, the authors cite Pan, Sezer and Roux, 2008 (and Fleetwood et al., 2020), but it is not clear that Pan, Sezer and Roux, 2008, did this, and in fact I think the first to do this from a swarms of trajectories solution was Lev et al., 2017 (see also Flood, et al., 2019), not cited here.

The reviewer is indeed correct here. We have updated the citations accordingly in the manuscript.

Having computed free energy projections along collective variables, receptor stability in relation to efficacy instead relies on estimation values for variables and not free energies, and the justifications for this could be improved. The authors write that the width and depth of the basins determine the most probable state projected into that space. This sentence is not clear, but its meaning is important, because ultimately the relative values of free energies are discussed. If the 2D maps are valid projections of the full configurational space (albeit with sampling guided/biased by the string), then Boltzmann integrals over each basin should yield a valid equilibrium constant. I presume the concern is that projections along different pairs of CVs can lead to different apparent free energies, because different CVs map out different proportions of phase space, and potentially envelop multiple states of the system, but does this mean a Boltzmann integral over a site is not a true thermodynamic quantity? In the Discussion, the authors return to add that examination of one projection can overlook what is happening in other coordinates/switches, but more discussion/justification is needed. Related, the authors state that they have "accurately captured the relative stability of states", but this is confusing given the relative stability of states from the maps appears to have been discredited.

We thank the reviewer for this astute set of comments, which we have enjoyed considering and addressing.

To illustrate our point, we highlight Figure 2 and Author response image 2, in which we can certainly say qualitatively that agonists stabilize the active basin. However, due to the issues with projections pointed out by the reviewer and the fact that these energy landscapes are not ideal two-state systems, an integration over state boundaries requires us to, somewhat arbitrarily, define the state boundaries. This will in turn influence any quantitative comparison to experimental values. The expectation values (Figure 2C-F), on the other hand, require no state boundaries, justifying our choice of this methodology.

**Author response image 2. respfig2:** 

That being said, we followed the reviewer’s advice and computed the relative stability of states for the different microswitches by taking the difference in free energy between the active and inactive regions of the free energy landscape (ΔG ∝-lnK, where K is the equilibrium constant). See the results in Figure 2—figure supplement 8 of the revised manuscript. State boundaries that were chosen are depicted as vertical dashed lines in Figure 2—figure supplement 7.The correlation between ΔG and cAMP signaling is strong for transmembrane microswitches, which also had a strong correlation between cAMP and their expectation values. Therefore, we see little practical difference between the two methods, but would recommend using the expectation value because it does not require defining arbitrary state boundaries.

Instead of using free energies, the authors correlate "expectation values" of their CVs to experiments in the bottom of Figure 2. I assume a plot against free energy was abandoned as it was not working as planned? Ideally one would want to see free energies plotted against experimental efficacy, because estimation values of variables such as TM5…, may correlate to efficacy, but not uniquely map to shifting state equilibrium.

Please see our comments in the previous paragraph. We chose expectation values since it does not require us to set state boundaries. This approach provides a single numerical value which, unlike having an entire probability distribution, is easier to correlate to an experimental numerical value. With that in mind, by merely taking the average and ignoring the characteristics of the free energy landscapes, such as the shape and locations of the basins, leaves out potential useful information. For this reason, we included the microswitch free energy landscapes as a basis of discussion together with the correlation plots in Figure 2.

Many analysis tasks were completed by existing packages for which the meanings are not obvious. e.g. Demystifying, Scikit-learn… Materials and methods, even if accepted, deserve a sentence to explain and motivate. e.g. CVs were short inverse inter-residue distances used to train a Restructured Boltzmann Machine (the principles of this machine could be explained simply). The subsection “Data driven analysis reveals that ligands stabilize unique states” is highly technical and not well explained. While many will know about PCA, less will be known about MDS and T-SNE. The ability to identify signaling hotspots in Figure 4 is impressive, leading to a model for allosteric communication. But the machine learning approaches used come across as black box and require better physical interpretation.

We thank the reviewer for pointing this out. It is easy to take one’s own methodological choice for granted! We have made revisions to clarify our rationale and better describe the tools used. For example, we describe what the software demystifying does when we mention it, and try to explain how we find important microswitches by computing the KL divergence in simpler terms.

Reviewer #2:This manuscript describes the use of enhanced sampling molecular dynamics to calculate free energy landscapes for the β_2_ adrenergic receptor. A particular focus is the conformational dynamics and thermodynamics of microswitches that change conformation upon receptor activation. A variety of ligands are investigated to identify shared and divergent features of receptor conformational modulation. Unbiased methods are shown to identify known conformational switches as key regulators of receptor activation.Overall the manuscript is interesting and clearly written. Figures are very clear and well presented. The subject matter has been very extensively studied in the GPCR family as a whole and in the β_2_ receptor in particular, and the results presented here largely align with existing understanding of GPCR activation and conformational regulation by ligands. A major concern is the question of whether the results presented here truly enhance our understanding of GPCR activation, or simply confirm things that are already known. Many groups have published detailed analysis of similar questions to those presented here, including the Dror, Nagarajan, and McCammon groups, among others. There is very little discussion of this prior work, which makes it difficult to see how the present manuscript fits within the broader context of GPCR activation molecular dynamics analysis.

We appreciate that the reviewer found the manuscript well presented. Following the reviewer’s comments, we have extended the discussion of prior work in the manuscript, as also described in the cover letter. Please also find arguments about the novelty of our work in the same place.

A significant technical concern is the reliance on cAMP Emax values as the sole experimental validation. A prospective experimental test of hypotheses generated here would significantly enhance the manuscript. Barring this, a more extensive comparison of computational results with experimental data is essential in my view. The cAMP pathway is highly amplified, which may confound analysis linking Emax values directly to conformational equilibria. Manglik et al., 2015, presents an actual biophysical analysis of conformational equilibria (albeit with fewer ligands) and comparison to this would be helpful. Do relative energy predictions match these experimental observations?

We believe that a comparison to cAMP measurements is relevant and interesting, since it represents a downstream cellular response and a strong correlation suggests our approach to have predictive powers. A comparison to Emax values instead of e.g. EC50 values makes sense, since our simulations have a ligand present at all times, which corresponds to experimental conditions with a very high ligand concentration. We agree that a prospective study would be ideal, but in the absence of experimental validation, we have followed the reviewer’s advice and have improved the comparison to experimental data, and in particular Manglik et al., 2015, as this was suggested by the reviewer. For a more detailed description of what has been done in this regard, refer to our comments under “Essential revisions” above.

A smaller issue is that the utility of the results presented here appears to be a bit overstated. For example, it is claimed (Introduction) that the results provide insight into how ligands with specific efficacy profiles can be designed. To support this statement, the authors should present examples of compounds they have designed based on their results, together with experimental data confirming these molecules have the intended efficacy profiles.

In the revised manuscript we are more precise in what our results allow us to do while avoiding speculative claims. We anticipate to use these methods in prospective studies going forward, but it is indeed not part of this study.

Reviewer #3:This is an interesting paper, and I like to attempt to connect analysis of allostery to function. However, I'm extremely concerned about statistical uncertainty - it's not really discussed, and it would be easy to chalk all of the results up to limited sampling. It will be important for the authors to demonstrate this isn't the case.

We are glad that the reviewer found our paper interesting and agree that the statistical validation was not clearly presented and also lacking specific tests.

Basically, my concern is that there's essentially no mention of statistical uncertainty or convergence anywhere in the document. One of the major claims is that the different agonists each populate a distinct substate - this is an incredibly important and interesting observation if true, but could also be explained by saying each simulation wandered in its own space and didn't have time to explore anywhere else. If it were run again, it might wander into a totally different place. I don't have a great feel for how rapidly swarms explore, but I do know the total sampling time of 1.4 microseconds per ligand sounds awfully small. There are examples in the literature showing that conventional simulations several times this length are not converged as far the configurations in the ligand binding pocket go (for example, Leioatts et al., Biophysical Journal, 2015, or some of the work from Dror's group).

The reviewer is right, we had omitted important information, relying on information present in our previous paper. We have revised the manuscript such that this manuscript now stands on its own. Specifically, our assessment of statistical reliability can be found in the cover letter.

Note that the single equilibrated states are by definition converged from a starting structure using an adaptive sampling approach with multiple walkers. Sampling a large region of conformational space would require much more than 1.4 microsecond, but a small region kinetically accessible from a starting structure may be well-sampled within this time-frame. Actually, an important methodological aspect of this manuscript is that we could identify ligand-induced pharmacological properties by adaptive sampling of a small portion of conformational space.

I have a couple of ideas for how the authors could convince me this isn't just a sampling artifact. The best would probably be to pick one ligand and redo the whole calculation 4 more times, start to finish - looking at the variation between those replicates would be a decent estimate of the uncertainty. Ideally, they'd do this for all of the ligands, but I recognize that's not a reasonable request, which is why I suggest doing it for 1 ligand.A weaker test would be to break the individual ligand calculations into blocks, somehow - I'm not totally sure how to do it - and show that the blocks are self-similar. If they're all wandering through the whole of the blob in Figure 3, you're more likely to be ok. If each block populates a discrete chunk of the blob (and the overall swarm data doesn't revisit), then there's a big problem.

We thank the reviewer for the specific suggestions! We indeed followed the reviewer’s second suggestion and performed stratified cross validation with 3 subsets of the walkers. More details can be found under Essential Revisions. We argue that this statistical test is as legitimate as rerunning the simulation for one ligand for two reasons. First, we can perform this test for all ligands, not just one, without wasting computational resources. Second, the method is based on finding a center point in CV space by tracking a number of walkers launched from the same starting structure. Considering the high number of walkers used and that our results pass a cross validation examination, we may conclude that the position of the center point is insensitive to the random seed of the individual MD trajectories. Thus, we expect a new set of walkers launched from the same starting structure to converge to the already sampled state.

I'd want to see error estimates on the free energies in Figure 2, plus a redo of the dimension reduction in Figure 3 to see if the ligands still separate more than the replicates of 1 ligand.

See our comments above.

On a more minor note, the use of p-values in the subsection “Ligands control efficacy by reshaping microswitches’ probability distributions”, isn't really correct. The point the authors are trying to make is that the computed value doesn't predict the experiment (for good reason), and the correlation coefficients make that point for you.

We have removed the use of the p-value and only refer to the correlation between cAMP response and microswitch expectation values.